# Infiltrating natural killer cells bind, lyse and increase chemotherapy efficacy in glioblastoma stem-like tumorospheres

Barbara Breznik [1,2✉], Meng-Wei Ko[1], Christopher Tse[3], Po-Chun Chen[1], Emanuela Senjor [4,5], Bernarda Majc [2,6], Anamarija Habič[2,6], Nicolas Angelillis[3], Metka Novak[2], Vera Župunski[7], Jernej Mlakar[8], David Nathanson[3] & Anahid Jewett[1✉]

Glioblastomas remain the most lethal primary brain tumors. Natural killer (NK) cell-based therapy is a promising immunotherapeutic strategy in the treatment of glioblastomas, since these cells can select and lyse therapy-resistant glioblastoma stem-like cells (GSLCs). Immunotherapy with super-charged NK cells has a potential as antitumor approach since we found their efficiency to kill patient-derived GSLCs in 2D and 3D models, potentially reversing the immunosuppression also seen in the patients. In addition to their potent cytotoxicity, NK cells secrete IFN-γ, upregulate GSLC surface expression of CD54 and MHC class I and increase sensitivity of GSLCs to chemotherapeutic drugs. Moreover, NK cell localization in peri-vascular regions in glioblastoma tissues and their close contact with GSLCs in tumorospheres suggests their ability to infiltrate glioblastoma tumors and target GSLCs. Due to GSLC heterogeneity and plasticity in regards to their stage of differentiation personalized immunotherapeutic strategies should be designed to effectively target glioblastomas.

[1] Division of Oral Biology and Medicine, The Jane and Jerry Weintraub Center for Reconstructive Biotechnology, University of California School of Dentistry, 10833 Le Conte Ave, Los Angeles, CA 90095, USA. [2] Department of Genetic Toxicology and Cancer Biology, National Institute of Biology, 111 Večna pot, 1000 Ljubljana, Slovenia. [3] Department of Molecular and Medical Pharmacology, David Geffen UCLA School of Medicine, 10833 Le Conte Ave, Los Angeles, CA 90095, USA. [4] Department of Biotechnology, Jožef Stefan Institute, 39 Jamova cesta, 1000 Ljubljana, Slovenia. [5] Faculty of Pharmacy, University of Ljubljana, 6 Aškerčeva cesta, 1000 Ljubljana, Slovenia. [6] The Jozef Stefan International Postgraduate School, 39 Jamova cesta, 1000 Ljubljana, Slovenia. [7] Chair of Biochemistry, Faculty of Chemistry and Chemical Technology, University of Ljubljana, 113 Večna pot, 1000 Ljubljana, Slovenia. [8] Institute of Pathology, Faculty of Medicine, University of Ljubljana, 2 Korytkova ulica, 1000 Ljubljana, Slovenia. ✉email: barbara.breznik@nib.si; AJewett@mednet.ucla.edu

Glioblastomas (gliomas WHO grade IV; GBMs) remain among the most aggressive malignancies of the brain, with no or little progress in the standard treatment for nearly 20 years. Only 5% of the patients survive 5 years or more and only half of the patients are still alive at 15 months after diagnosis[1,2]. Cancer cells with self-renewing properties, named GBM stem-like cells (GSLCs), propagate GBMs and express stemness markers[3,4]. GSLCs are resistant to standard therapy due to efficient DNA repair mechanisms and multidrug resistance causing regrowth of GBM tumors[4,5]. Moreover, GSLC ability to evade immunosurveillance[4] and immunosuppressive GBM microenvironment prevent efficiency of checkpoint blockade and CAR T cell approaches in GBM therapy[6,7].

Natural killer (NK) cells are large granular and cytotoxic lymphocytes that are important first-line defense immune effectors against tumors. NK cells are able to recognize and spontaneously kill damaged and stressed infected or tumor cells, without prior sensitization. NK cell activity and interaction with target cells are mediated by a balance between stimulatory and inhibitory receptors on NK cell surface and ligands on the target cells[8,9]. For instance, tumor cell surface presence of major histocompatibility complex molecules (MHC) class I, which interact with inhibitory killer immunoglobulin-like receptors on NK cells, induces inhibitory signals which may allow the survival of tumor cells[9–11]. NK cells have ability to control cancer progression directly via cancer cell lysis and indirectly by regulating innate and adaptive antitumor immune responses. NK cell killing of target (tumor) cells is mediated through secreted granules containing membrane-disrupting proteins perforins and proteolytic enzymes granzymes that trigger target cell lysis. NK cell killing can also be mediated by factors from the tumor necrosis factor (TNF) family[9–12].

NK cell-based immunotherapy is a perspective approach in treatment of several cancers, including GBMs[9,13,14]. Namely, NK cells have been shown as the only immune effectors known to recognize and kill GSLCs, without approaches to generate immunogenic antigens and cell priming with appropriate costimulatory signals, as are needed for potential T cell- or dendritic cell (DC)-based immunotherapies[3,11,15–17]. This is important in GBMs since these tumors are extremely heterogeneous at the molecular and cellular levels[1], expressing or lacking specific antigens in subsets of tumor cells, thereby exhibiting different genetic fingerprints in a single tumor. In addition, tumor-infiltrating immune cells, such as macrophages and Tregs, generate immunosuppressive microenvironment[6], lacking antigen-presenting potential and costimulatory antigens, leading to the tumor resistance to immunotherapy.

NK cells that can select and kill cancer stem-like cells are CD16$^{bright}$CD56$^{dim}$CD69$^-$ population and represent 90% of peripheral blood NK cells[11]. In tumor microenvironment upon interactions with tumor cells and stromal cells, NK cells lose CD16 expression and increase expression of CD56 surface receptor, leading to decreased cytotoxicity and increased production of cytokines, including interferon gamma (IFN-γ). These cells are considered to be split anergized NK cells and are able to regulate the function of other cells, tumor and immune, in the tumor microenvironment[11,18,19]. Various cytokines, including IL-2, IL-15 and IL-12, can be used to activate and expand NK cells in vitro[20]. It has been shown previously that split anergy in NK cells can be achieved by the treatment of NK cells with human recombinant IL-2 and anti-CD16 monoclonal antibodies[10,18,21].

In addition to cytotoxic function, NK cells act as regulatory cells and secrete various pro- and anti-inflammatory cytokines and chemokines, such as IFN-γ and interleukin (IL)−6 that orchestrate innate and adaptive immune responses and shape tumor microenvironment. For example, NK cells boost the tumor infiltration as well as maturation and activation of DCs and T cells and by that promote antitumor immune responses[12,22]. NK cells provide critical signals to tumor cells, which results in the differentiation of cancer stem-like cells[18]. IFN-γ, secreted from split anergized NK cells, was shown to be primarily responsible for NK-mediated increase in tumor cell surface expression of MHC class I, CD54, which is intercellular adhesion molecule I (ICAM1), and programmed cell death receptor ligand 1 (PD-L1), a ligand for immune checkpoint protein PD-1[10,18,23]. Those surface markers, together with CD44, are associated with immune cell function, such as resistance or susceptibility to NK cell and T cell-mediated recognition and lysis[10,18,19,23–25], as well as differentiation stage of tumor cells[10,11,18,19,23,26]. Due to their crucial role in antitumor immune responses, NK cells and NK cell-directed immunotherapies are currently in several preclinical and clinical studies for treatment of hematological and solid malignancies. These studies have shown encouraging clinical responses and have been shown to be safe in cancer patients. Despite the great potential of NK cell-based therapies, there are still many challenges to translate the use of NK cells into the clinical practice. These include prolonging the persistence of NK cells in vivo and overcoming their exhaustion, as well as in vitro expansion to obtain sufficient quantity of efficient therapeutic NK cells[27].

Several in vitro NK expansion techniques have been developed to allow for a higher therapeutic cell dose. Super-charged NK cells are highly activated cytotoxic NK cells with high potential for cytotoxicity and secretion of cytokines, recently developed by our group for adoptive NK cell-transfer therapy. Super-charged NK cells are produced using sonicated probiotic bacteria AJ2 and osteoclasts as feeder cells that provide all necessary signals to activate and expand NK cells, including cytokines IL-12 and IL-15[19,23].

In the present study, we aimed to investigate the therapeutic potential of allogeneic highly activated super-charged NK cells in GBM tumors by studying the interactions between NK cells and patient-derived GSLCs. We tested the therapeutic effect of NK cells by analyzing NK-mediated cytotoxicity and secretion of pro-inflammatory cytokines in 2D and 3D in vitro GBM models. In order to study NK cell penetration and infiltration in GBM tumors in vivo, NK cell markers immunostaining of GBM tissue sections and NK cell penetration studies into GSLC tumorospheres were performed. We aimed to characterize the differential susceptibility of GSLCs to NK-mediated lysis with respect to their phenotypic characteristics in vitro and in vivo. The NK-mediated GSLC phenotypic changes and sensitivity of GSLCs to chemotherapeutic drugs, including temozolomide (TMZ), used in the standard GBM treatment protocols[28], were also analyzed.

## Results

**NK cell markers were present in peri-vascular GBM tumor regions in close proximity of GSLC markers, and NK cells are functionally inactivated in peripheral blood of GBM patients.** NK cell marker CD56-positive cells were present in tissue sections in 6 out of 8 GBM tumors. However, the abundance of CD56 immunostaining through GBM tissues was low. CD56-positive cells were detected in specific areas around the vasculature (Supplementary Fig. 1). In the majority of samples, CD56-positive cells were detected in close proximity to SOX2-positive GBM cells and SMA-positive vascular cells and large vessels—arterioles. CD56-positive cells were in direct cell-cell contact with cells, positive for GSLC markers SOX2 (Fig. 1a, b, e) and CD44 (Supplementary Fig. 2). CD3 staining was used to determine specificity and exclude NKT cells (Fig. 1c). GFAP and CD44 staining was performed to confirm that CD56-positive cells

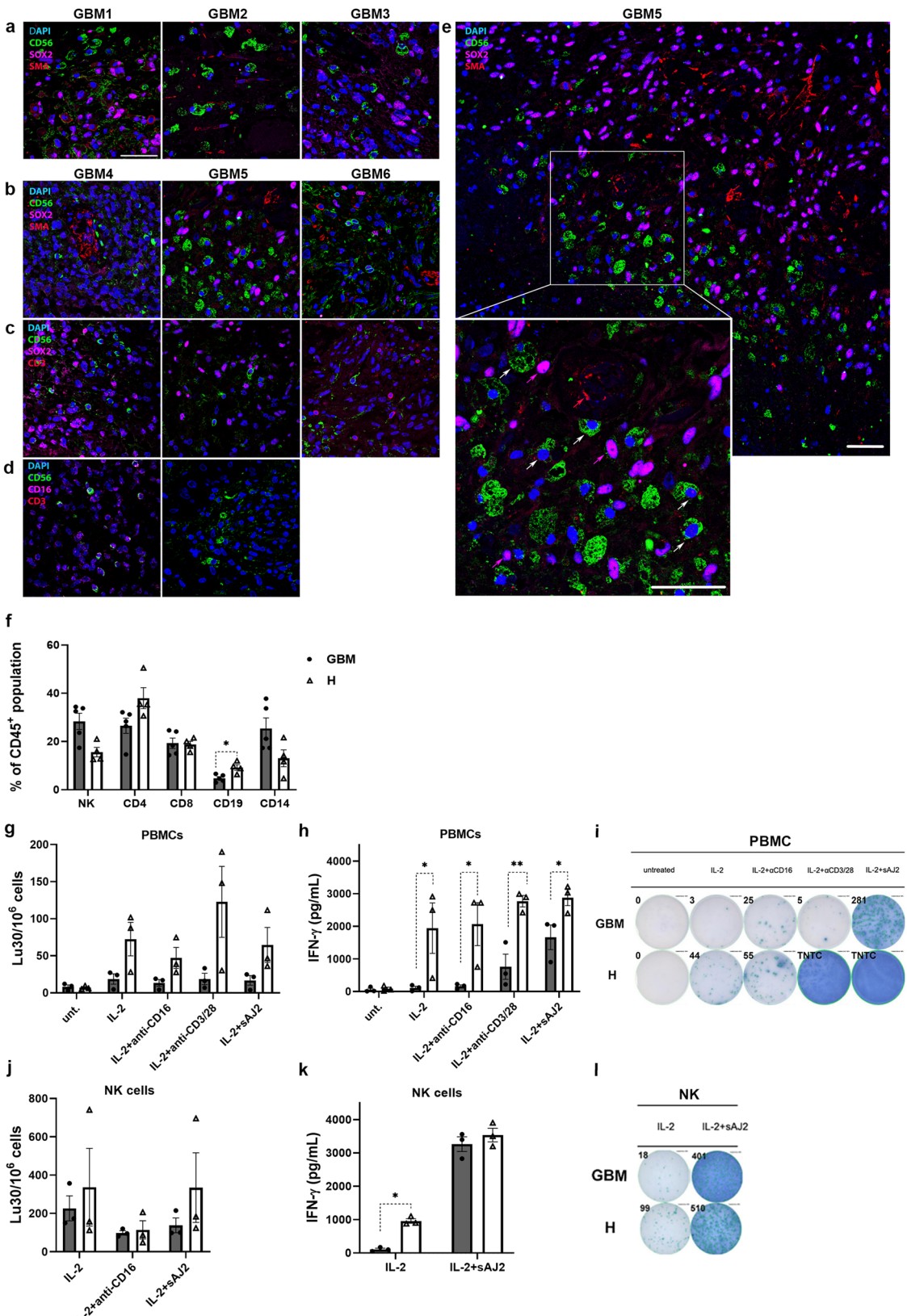

are not GBM cells (Supplementary Fig. 2). CD45 staining was performed to confirm that the majority of CD56-positive cells are indeed immune cells (Supplementary Fig. 3). CD56-positive cells were negative for cytotoxic T cell marker CD8 and activating NK cell receptor NCR1 (NKp46). NCR1 was detected only in 1 GBM tissue (Supplementary Fig. 4). CD16 immunostaining of NK cells was also performed, and although a few CD56$^{dim}$ and CD16$^{+}$ subsets of NK cells could be observed, it appeared that CD56$^{+}$CD16$^{−}$ NK cells were more dominant in GBM tissues (Fig. 1d).

NK cell number was assessed in the peripheral blood of five GBM patients as compared to four healthy donors. NK cell function was assessed in the peripheral blood of three GBM

**Fig. 1 NK cell markers presence in peri-vascular tumor regions and NK functional inactivation in peripheral blood of GBM patients.** SOX2-positive cells (indicated by magenta arrows) and CD56-positive cells (indicated by white arrows) are found together around SMA-positive vasculature (red arrows) (**a**, **b**, **e**). CD56-positive NK cells were negative for T cell marker CD3 (**c**). NK markers CD56 and CD16 co-localized in 1 GBM tissue (**d**). Assessment of immune cell percent within CD45-positive population in peripheral blood of GBM patients and heathy donors was determined by immunolabeling and flow cytometry (**f**). NK cell function assessment of GBM patient and healthy donor PBMCs (**g–i**) and NK cells (**j–l**) after several treatments: IL-2, IL-2 + anti-CD16, IL-2 + anti-CD3/28 and IL-2 + aAJ2 as described in Methods section. NK cell cytotoxicity assay (**g**, **j**), ELISA (**h**, **k**) and ELISPOT (**i**, **l**) for IFN-γ secretion were performed. Cytotoxic ability of patient-derived primary NK cells was tested on UC2 cells. Data are showed for 4 heathy donors (white bars) and 5 GBM patients (gray bars) (**f**), and for 3 healthy donors and 3 GBM patients (**g–l**), respectively. Data are presented as means ± SEM, data points represent measurement for each healthy donor/patient Scale bar = 50 μm. TNTC: too numerous to count.

patients as compared to three healthy donors (Fig. 1f-l). NK cell numbers were not altered in GBM patients (Fig. 1f). We detected decreased NK cell function in GBM patients, such as decreased NK cell-mediated cytotoxicity in PBMCs and NK cells (Fig. 1g, j) and lower IFN-γ secretion by PBMCs (Fig. 1h, l) as well as NK cells (Fig. 1k, l) when compared to those of healthy donors.

**GSLCs exhibited different susceptibility to NK cell-mediated lysis and expressions of cell surface CD44, CD54, MHC class I and PD-L1.** We analyzed several GBM cells for NK cell-mediated lysis. GS025 GSLCs were least sensitive and NCH421k GSLCs the most sensitive to primary allogeneic NK cell-mediated lysis. Oral squamous carcinoma stem-like cells (UC2) demonstrated the highest susceptibility to NK cell lysis previously[26] and were used as the positive control (Fig. 2a). Split anergized NK cells, which were IL-2 and anti-CD16 monoclonal antibody (mAb) pre-treated, exhibited lower lysis of the GSLCs when compared to IL-2 activated (cytotoxic) NK cells (Fig. 2a). GSLCs were analyzed for the cell surface expressions of immune-related markers[11,23,26] (Fig. 2b). CD44 and CD54 were expressed only by GS025 cells. In contrast, MHC class I and PD-L1 were expressed on the cell surface of both tested GSLCs. GS025 cells expressed higher levels of MHC class I in comparison to the NCH421k cells.

Since NK cells are known to release cytokines, notably IFN-γ, we tested the effect of recombinant IFN-γ on the expression of the above-mentioned markers on the surface of GBM cells. IFN-γ increased the expression of CD54 and MHC class I on GS025 cells, which were also found to be less susceptible to NK cell-mediated lysis. In NCH421k cells, which were highly susceptible to NK cells, IFN-γ only increased the expression of MHC class I (Supplementary Fig. 5). Based on their differential susceptibility to NK-mediated cytotoxicity and expression of cell surface markers before and after IFN-γ treatment, we selected two model GSLCs with the lowest vs. highest response to NK cells, GS025 and NCH421k, respectively.

**Highly proliferative and tumorigenic NCH421k cells were more susceptible to primary and super-charged NK cell cytotoxicity compared to slowly proliferating GS025 cells.** NCH412k and GS025 cells were tested against primary and super-charged NK cell-mediated cytotoxicity in 2D cultures. Similar to the previous experiments, GS025 cells were moderately affected by primary NK cells, whereas a 4-fold increase in killing was observed with super-charged NK cells. NCH421k cells were significantly more susceptible to primary NK cell-mediated lysis, but only slightly more (10%) to super-charged NK cell killing, likely due to the plateau effect of NK cell cytotoxicity in chromium release assay. Split anergized NK cells killed NCH421k tumors significantly less than IL-2 activated primary NK cells. However, split anergized super-charged NK cells had higher killing ability compared to split anergized primary NK cells (Fig. 2c).

GS025 and NCH421k cells expressed similar protein levels of astrocyte marker glial fibrillary acidic protein (GFAP). Serum components induce GSLC differentiation and expression of GFAP. After exposure of GS025 and NCH421k cells to 1% FBS, GFAP expression was significantly increased in GS025 cells, but there was no change in GFAP expression in NCH421k cells (Fig. 2d).

Cell proliferation of both GSLCs was compared in vitro and in immune-deficient NSG mice. NCH421k cells proliferated significantly faster in cell cultures and in tumorospheres compared to GS025 cells (Fig. 2e). After intracranial injection, larger tumor growth was observed within 25 days when NCH421k cells were implanted into the right hemisphere of the mouse brain than when GS025 cells were injected (Fig. 2f). In vivo measured tumor burden and very fast growth after isolation from mouse brain confirmed that NCH421k cells were more proliferative compared to GS025 cells, which were not detected after isolation (Fig. 2g, h).

**Cytotoxicity of super-charged NK cells was much higher than that of primary NK cells and was mediated by direct cell-cell interactions in the 3D GBM model.** Confocal microscopy and single z-stack planes showed that primary and super-charged NK cells penetrated into the tumorospheres (Fig. 3a). 3D spot and surface rendering analysis of specific areas revealed direct contact between NK cells and GSLCs, indicating direct interactions of NK cells with the GSLCs in the tumorospheres (Fig. 3b–d). Primary and super-charged NK cells decreased the numbers of GSLCs and increased the number of dead cells in GSLC tumorospheres. Super-charged NK cells exhibited much higher killing than primary NK cells (Fig. 4a, b). Morphologically different GSLC tumorospheres were observed after addition of NK cells. GS025 GSLCs were more dispersed and mixed with NK cells in the culture wells, whereas NCH421k tumorospheres cultured with primary NK cells retained a compact spherical structure.

The killing effect of NK cells was quantitated by flow cytometry. Super-charged NK cells killed significantly more GS025 and NCH421k cells than primary NK cells and split anergized super-charged NK cells. The killing effect of super-charged NK cells was higher in NCH421k tumorospheres (Fig. 4c).

**Increased IFN-γ secretion in 2D and 3D cultures of NK cells with GSLCs.** Primary NK cells from the healthy donors used in the cytotoxicity assay were used to determine IFN-γ secretion. Significantly increased secretion of IFN-γ was detected in all ratios of GSLCs co-cultured with primary NK cells than in NK cell monocultures. Increased secretion of IFN-γ was also demonstrated in split anergized NK cells when compared to IL-2 activated NK cells. In monocultures of GSLCs, IFN-γ secretion was under the limit of detection. NK cells secreted significantly more IFN-γ in the presence of NCH421k than in the presence of GS025 cells (Fig. 5a). The number of IFN-γ secreting NK cells was not altered or even decreased in the co-cultures of primary NK cells in different cell ratios with GS025 cells. The trend was the

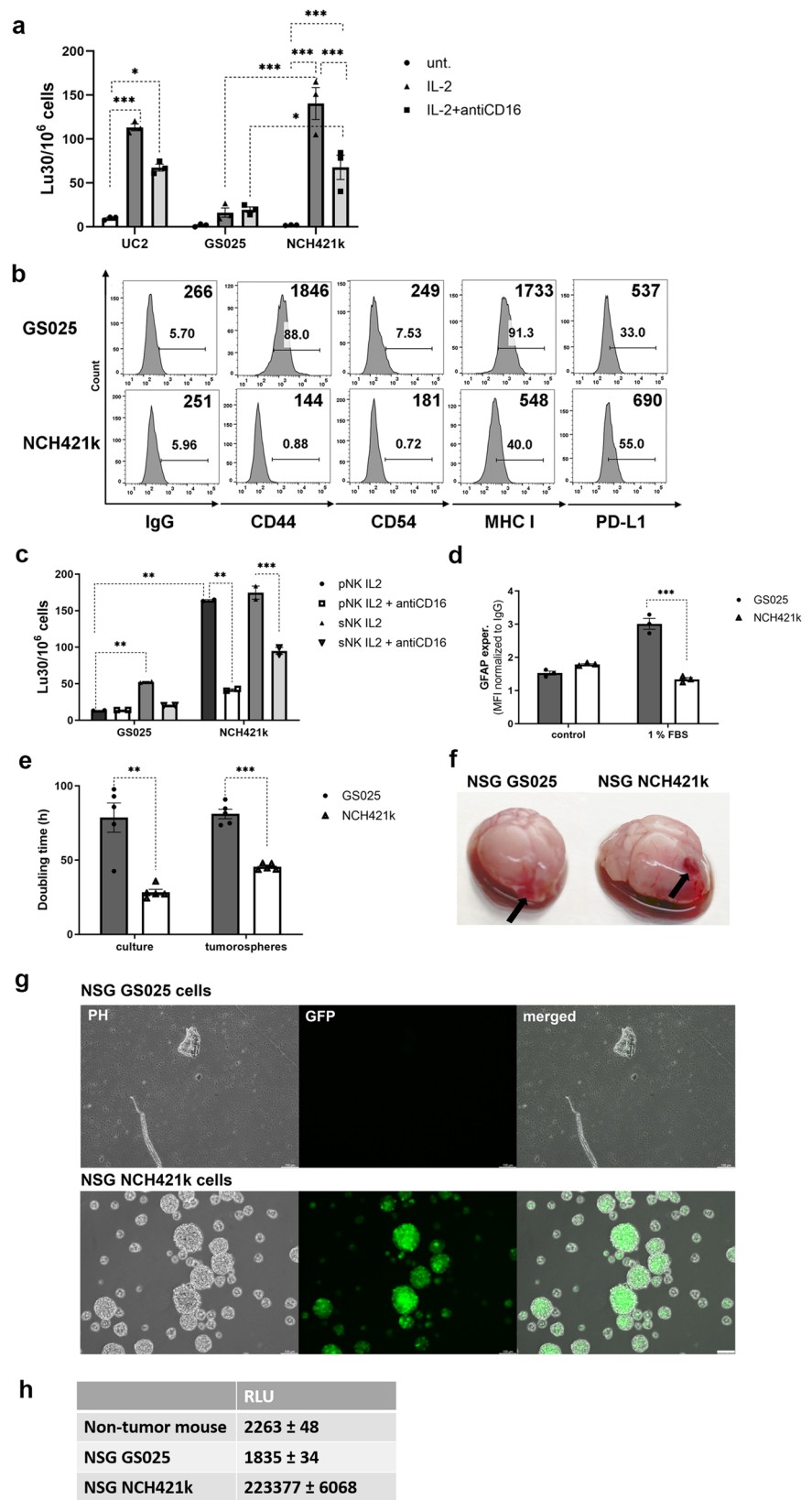

opposite in NCH421k and NK cell co-cultures, where increased numbers of IFN-γ secreting NK cells was noticed (Fig. 5b).

Similarly, the secretion of IFN-γ was increased in NCH421k tumorospheres with super-charged NK cells, when compared to those infiltrated in GS025. In addition to INF-γ, the levels of other secreted chemokines and cytokines were higher in NCH421k tumorospheres with super-charged NK cells, when compared to those infiltrating GS025 (Supplementary Fig. 6).

**Fig. 2 Primary and super-charged NK cells killed significantly more fast growing and tumorigenic NCH421k GSLCs than slowly growing GS025.** Oral squamous carcinoma stem-like cells (UC2) and GSLCs were tested for NK cell-mediated lysis. NK cells, isolated from healthy donors ($n = 3$, data points represent measurement for each healthy donor) were used. Healthy donor NK cell cytotoxicity was assessed after treatments: untreated (whiter bars), IL-2 (dark gray bars) and IL-2 + anti-CD16 mAb (light gray bars) (**a**). Cell surface marker expression was analyzed by flow cytometry (**b**). GS025 and NCH421k GSLCs were tested for primary and super-charged NK cell-mediated lysis. Primary and super-charged NK cells originated from different donors ($n = 2$, data points represent measurement for each healthy donor). NK cytotoxicity was assessed after treatments: IL-2 (black and dark gray bars for pNK and sNK, respectively) and IL-2 + anti-CD16 mAb (white and light gray bars for pNK and sNK, respectively) (**c**). GFAP protein expression was analyzed in GS025 (gray bars) and NCH421k (white bars) GSLCs after their exposure to 1% FBS for 6 days or control (neurobasal cell culture medium) using flow cytometry ($n = 3$, independent biological replicates) (**d**). GS025 (gray bars) and NCH421k (white bars) GSLC proliferation was evaluated by doubling time measurement in cell cultures and tumorospheres ($n = 5$, independent biological replicates) (**e**). NSG mice brain with GSLC tumor (as shown by arrows) 25 days after GSLC injection are shown (**f**). Isolated GFP-transduced GSLCs from mice (**g**). Tumor burden was assessed by secreted *Gaussia* luciferase activity (relative light units) in mice peripheral blood of non-tumor and GBM-bearing NSG mice ($n = 3$) (**h**). Data are presented as means ± SEM. Scale bar = 100 μm.

**Increased levels of secreted IL-6 in 2D co-cultures of primary NK cells with GSLCs.** Significantly increased secretion of IL-6 was detected in co-cultures of primary split anergized NK cells with GSLCs as compared to primary split anergized NK cells alone. Higher IL-6 increase was seen with split anergized NK cells in co-culture with NCH421k cells in comparison to those with GS025 cells. Split anergized NK cells secreted more IL-6 compared to IL-2 activated NK cells in co-cultures with GSLCs (Fig. 5c).

**Increased IL-6 and IL-8 levels in cerebrospinal fluid of GBM patient.** We next determined the different subpopulations of cerebrospinal fluid (CSF) mononuclear cells and the secretion of cytokines and chemokines in CSF of GBM patient. T cells were the most represented, with CD8 + T cells being higher than the CD4 + T cells. There were almost equal percentages of NK cells, B cells and monocytes (Supplementary Fig. 7). In untreated CSF fluid, we detected high levels of IL-6 and IL-8 followed by MIP-1α and MIP-1β (Supplementary Fig. 8a). Because of higher representations of T cells we treated CSF with IL-2 or IL-2 and anti-CD3/CD28 mAb to activate T cells, and determined the levels of secreted cytokines and chemokines. The levels of IL-6 were similar after both treatments and were much lower in the untreated CSF. However, the levels of other high expressing cytokines and chemokines GM-CSF, MIP-3α, IL-8, TNF-α and others increased in treated CSF and the highest levels of those cytokines were seen in CSF treated with IL-2 and anti-CD3/CD28 mAb. Some of the lower expressing cytokines were also elevated in treated CSF when compared to untreated samples (Supplementary Table 1). Most notably IFN-γ was elevated in treated samples and when the ratios of IL-6 to IFN-γ and IL-8 to IFN-γ were assessed, they were much lower in IL-2 and anti-CD3/28 mAb treated samples (Supplementary Fig. 8b, c).

**NK supernatants and the chemotherapeutic agent temozolomide modulated surface expressions of CD44, CD54, MHC class I and PD-L1 on GSLCs.** NK cell supernatants increased cell surface expressions of CD54 and MHC class I in GSLCs in cell cultures, whereas surface expressions of CD44 and PD-Ll were not altered (Fig. 6a). Similarly, expression of MHC class I on the surface of GSLCs in tumorospheres was increased after the addition of super-charged NK cells and the levels remained higher in GS025 cells than in NCH421k cells (Fig. 6b). NK supernatant did not alter the expression of astrocytic differentiation marker GFAP in GSLCs (Supplementary Fig. 9).

When TMZ was added to the tumor cells there was an upregulation of CD44, CD54 and MHC I expression in NK supernatant-treated GS025 and NCH421k cells as compared to only NK supernatant- and only TMZ-treated cells. PD-L1 surface expression was increased only in GS025 cells after NK

supernatant and TMZ treatment. The levels of increase in surface marker expressions were much higher in GS025 cells than in NCH421k (Fig. 6a).

**NK supernatants increased chemotherapeutic sensitivity of GSLCs.** Combined treatment with NK supernatants and chemotherapeutics of GS025 and NCH421k cells decreased GSLC cell number when compared to treatment with chemotherapeutics alone (Fig. 6c–f). With respect to cell death cisplatin (CDDP) and TMZ induced higher death of NCH421k cells, treated with NK supernatants as compared to control cultures, treated only with chemotherapeutics. However, NK cell supernatants alone, without addition of TMZ, already increased cell death of NCH421k cells. There was no additional increase in cell death after exposure of GS025 cells to combined treatment (Fig. 6g–k).

**Serum differentiation and IFN-γ treatment of GSLCs impaired super-charged NK cell-mediated lysis and were associated with altered levels of ligands for NK cell activating and inhibitory receptors, as well as decreased levels of OLIG2.** To gain mechanistic insight into super-charged NK cell-mediated lysis of GSLCs, NCH421k cells were exposed to serum differentiation (10% FBS) for 7 days[3] and to IFN-γ for 48 h. The NK cell-mediated lysis of GSLCs was then monitored. Subsequently, GSLC proliferation and GSLC protein expression of several ligands for activating and inhibitory NK cell receptors, proteins associated with immune cell functions, GSLC and GBM markers were evaluated. Treatment of NCH421 cells with 10% FBS and IFN-γ resulted in a decrease in super-charged NK cell cytotoxicity by 65 and 60%, respectively (Fig. 7a). NCH421k cells treated with 10% FBS and IFN-γ were less proliferative and expressed lower protein levels of the GSLC marker OLIG2 (Fig. 7a–c). The reduced cytotoxicity of super-charged NK cells in cells treated with 10% FBS was associated with lower levels of activating NK cell ligands (CD112, CD155) compared with the NCH421k control. In IFN-γ-treated cells, decreased cytotoxicity of super-charged NK cells was associated with decreased levels of the activating NK cell ligand CD112, increased levels of the activating NK cell ligands CD155 and ULBP2/5/6, and increased levels of the inhibitory NK cell ligand MHC class I compared with the NCH421k control. Cells treated with 10% FBS expressed less OLIG2, CD155, ULBP2/5/6, ULBP3, and MHC class I compared with IFN-γ-treated cells (Fig. 7c, Supplementary Figs. 11–13). Other ligands of activating (B7-H6, ULBP1, MICA /B) and inhibitory (HLA-E, PD-L1, CD54) NK cell receptors were either expressed in lower levels or not expressed in NCH421k cells. The expression of the astrocytic marker GFAP and the GSLC marker CD133 was not changed in NCH421k cells after the treatments (Supplementary Figs. 10–13).

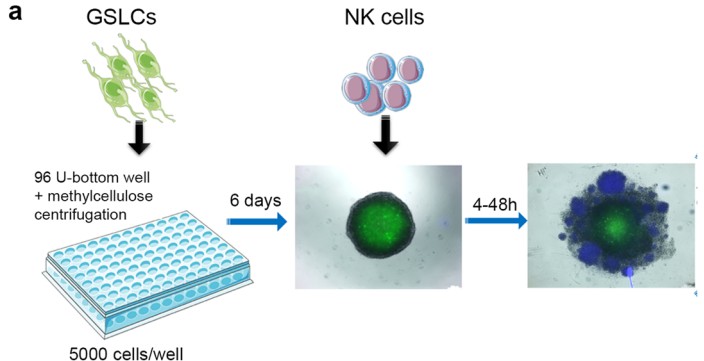

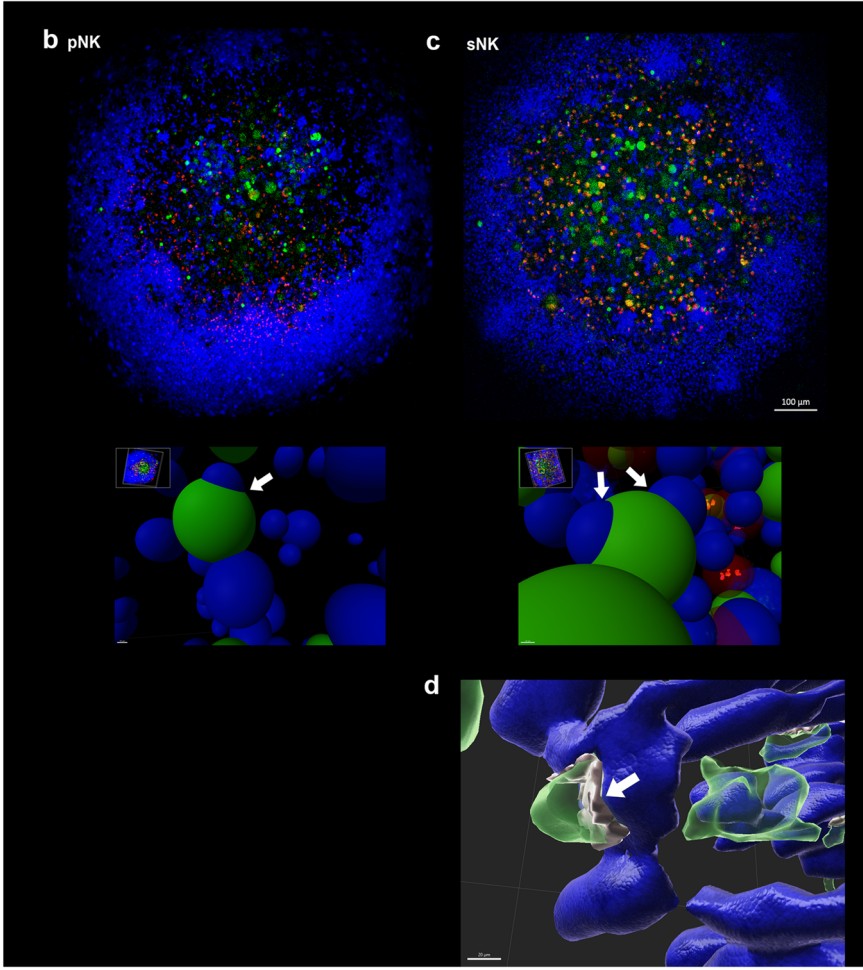

**Fig. 3 NK cells (blue) penetrated into the GSLC tumorospheres (green) and directly interacted with tumor cells.** IL-2 activated primary NK cells (**a**, **b**) or super-charged NK cells (**c**) were added to NCH421k tumorospheres in ratio 2.5:1 for 24 h before images were taken by confocal microscope. Single focal plane ($z = 33\,\mu m$) showed penetration of NK cells into the GSLC spheroid and co-localization of blue and green cells within spheroids (*upper* **b**, **c**). Spot analysis in Imaris software of selected areas, where GSLC and NK cell signals co-localized (arrows), revealed direct NK-GSLC interactions (*lower* **b**, **c**). Surface rendering of GSLCs and NK cells in Imaris showed direct cellular interactions (**d**). Gray surfaces represents areas of green GSLC surfaces touching blue surface of NK cells (arrow). Red dots represent PI-positive dead cells. Scale bar = 100 µm (upper **b**, **c**), 10 µm (lower **b**, **c**) and 20 µm (**d**).

## Discussion

NK cell-based immunotherapy is a promising antitumor treatment strategy as NK cells have been shown to selectively kill cancer stem-like cells[10,29]. It has been proposed recently that NK cells also exhibit a crucial role in the control of metastasis by eliminating circulating cancer cells with stem-like cell characteristics[8]. However, during tumor progression, cancer cells acquire an immune resistance and create immunosuppressive tumor microenvironment, partially counteracting the NK cell functions[30]. For example, it has been demonstrated that NK cell activity is compromised in peripheral blood of some tumor patients[19,24]. Similarly, we detected decreased NK cell-mediated cytotoxicity and IFN-γ secretion in peripheral blood of GBM patients when compared to NK cells from healthy donors.

In the present study, NK cell markers were detected in human GBM tissue sections in close proximity of tumor vasculature and cells positive for GSLC markers SOX2 and CD44. GBM tissue regions, enriched with GSLCs and tumor vasculature, are called GSLC niches and were shown to associate with GSLC maintenance and protection, increasing resistance against standard

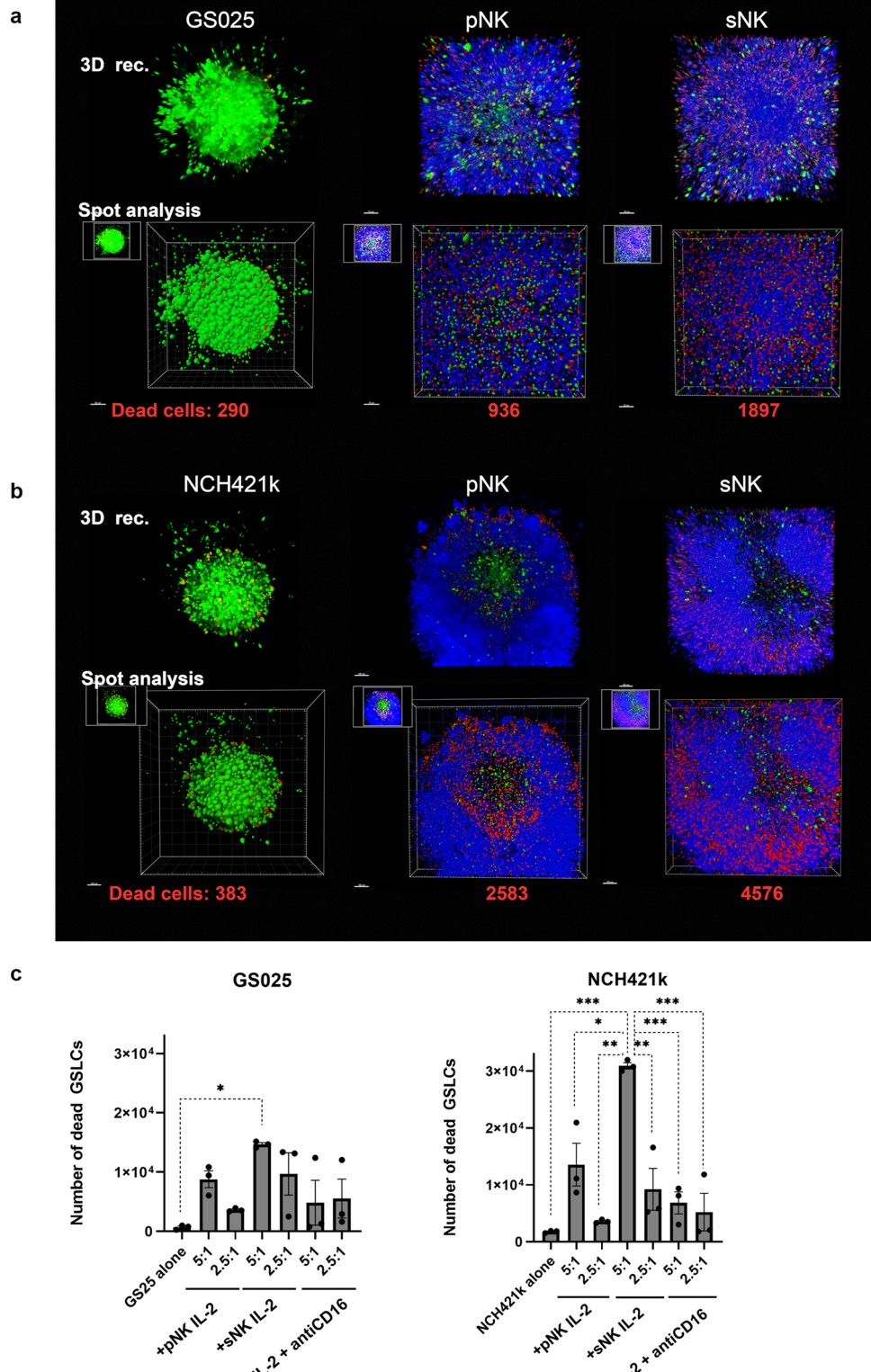

**Fig. 4 Primary and super-charged NK cells triggered GSLC death and decreased the number of GSLCs in spheroids.** Primary (p) and super-charged (s) NK cells (blue) were treated (IL-2 or IL-2 + antiCD16 mAb) and added to GS025 (**a**, *left* **c**) and NCH421k (**b**, *right* **c**) tumoropheres (green) in different NK: GSLC ratios (5:1 and 2.5:1) for 48 h. 3D reconstructed images in blend mode and spot analysis images, are presented in upper and lower rows for each GSLCs after, respectively. NK cells were added to GSLC tumoropsheres in 5:1 ratio. Images were taken after 24 h of co-culture. The number of dead GSLCs per spheroid (**c**) were determined 48 h after the addition of NK cells. Data are presented as means ± SEM ($n = 3$, data points represent measurement for each healthy donor). Scale bar = 70 μm (**a**), 100 μm (**b**).

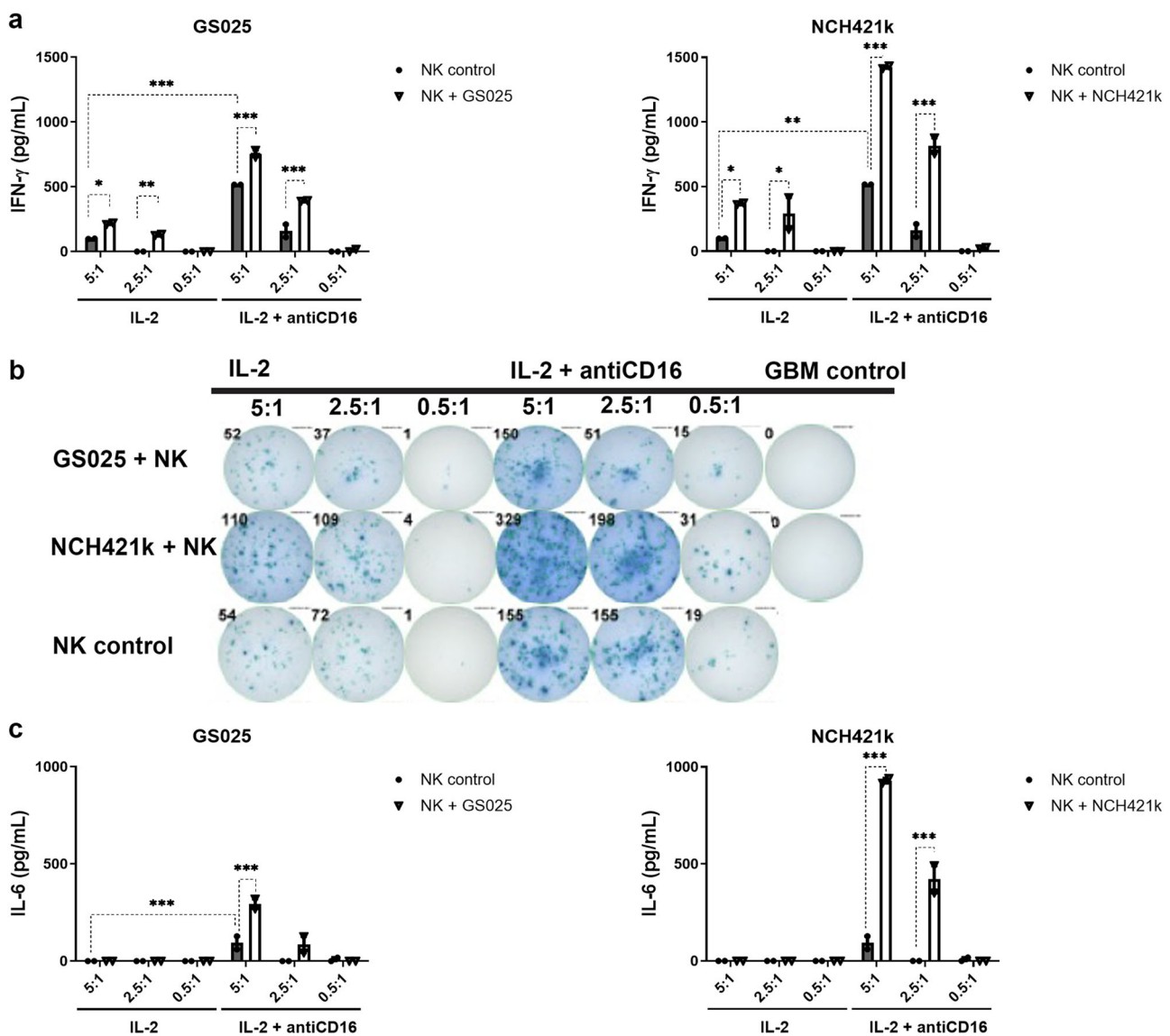

**Fig. 5 Increased IFN-γ and IL-6 secretion in 2D co-cultures of NK cells and GSLCs in different rations (5:1, 2.5:1, 0.5:1).** ELISA (**a**, **c**) and ELISPOT (**b**) were used to investigate IFN-γ and IL-6 secretion and number of blue IFN-γ secreting NK cells (**b**) after 24 h of co-culture, respectively. Gray bars represent NK cell monocultures (controls) and white bars represent co-cultures of NK cells and GSLCs. Data are presented as means ± SEM, data points represent independent biological replicates (*n* = 2).

therapeutic approaches[1,31]. In previous studies, we demonstrated that GSLC niches are hypoxic and peri-vascular[32,33]. GSLCs are localized adjacent to the tumor vasculature in hypoxic areas in GBM tumors, where hypoxia is one of the prime conditions for stem cell maintenance[32]. In the majority of patient-derived GBM tissues, NK cell markers are present in close proximity to peri-vascular niches, indicating that these cells can cross the blood-brain barrier (BBB) and reach GSLCs. This notion supports previous studies on NK cells which demonstrated penetration through the BBB, and chemotactic infiltration to the tumor sites[34–36]. Our immunohistochemical data also demonstrated that a certain population of NK cells, expressing CD56 in some tumors did not express CD16 receptor, implicating a possible down-regulation of CD16 by GSLCs or by other stromal elements in the niches. Lack of CD16 expression in tumor-residing NK cells is in line with previous reports on decreased CD16 expression after NK cell interactions with cancer stem-like cells[34], demonstrating impairment of NK cell cytotoxicity but increasing IFN-γ secretion. This NK cell state is annotated as split anergized NK cells[11,18].

NK cell delivery to GBM is considered a perspective therapeutic approach but large-scale production and activation of NK cells are still challenging[9,20]. Our previous work has demonstrated that osteoclasts as a feeder layer and probiotic bacteria can stimulate growth and activation of primary peripheral blood-derived NK cells to become super-activated and as such highly cytotoxic cells with high secretion of cytokines up to 1 month in cell cultures after their isolation from healthy donors. These cells are named super-charged NK cells[19]. One example of activating signals that are provided by osteoclasts are IL-15 and IL-12. Namely, IL-15 blocking resulted in impaired cytotoxicity and expansion of super-charged NK cells in vitro. NK cell-mediated cytotoxicity and IFN-γ secretion were decreased in super-charged NK cell cultures after IL-12 blocking[19]. These super-charged NK cells prevented growth and tumor-initiating potential of pancreatic and oral tumor stem-like cells as well as restored immune cell effector functions in humanized BLT mice[19,23,26].

We showed that super-charged NK cells kill GSLCs in in vitro experiments in 2D co-cultures and 3D tumorospheres. In comparison to IL-2 activated primary NK cells, super-charged NK

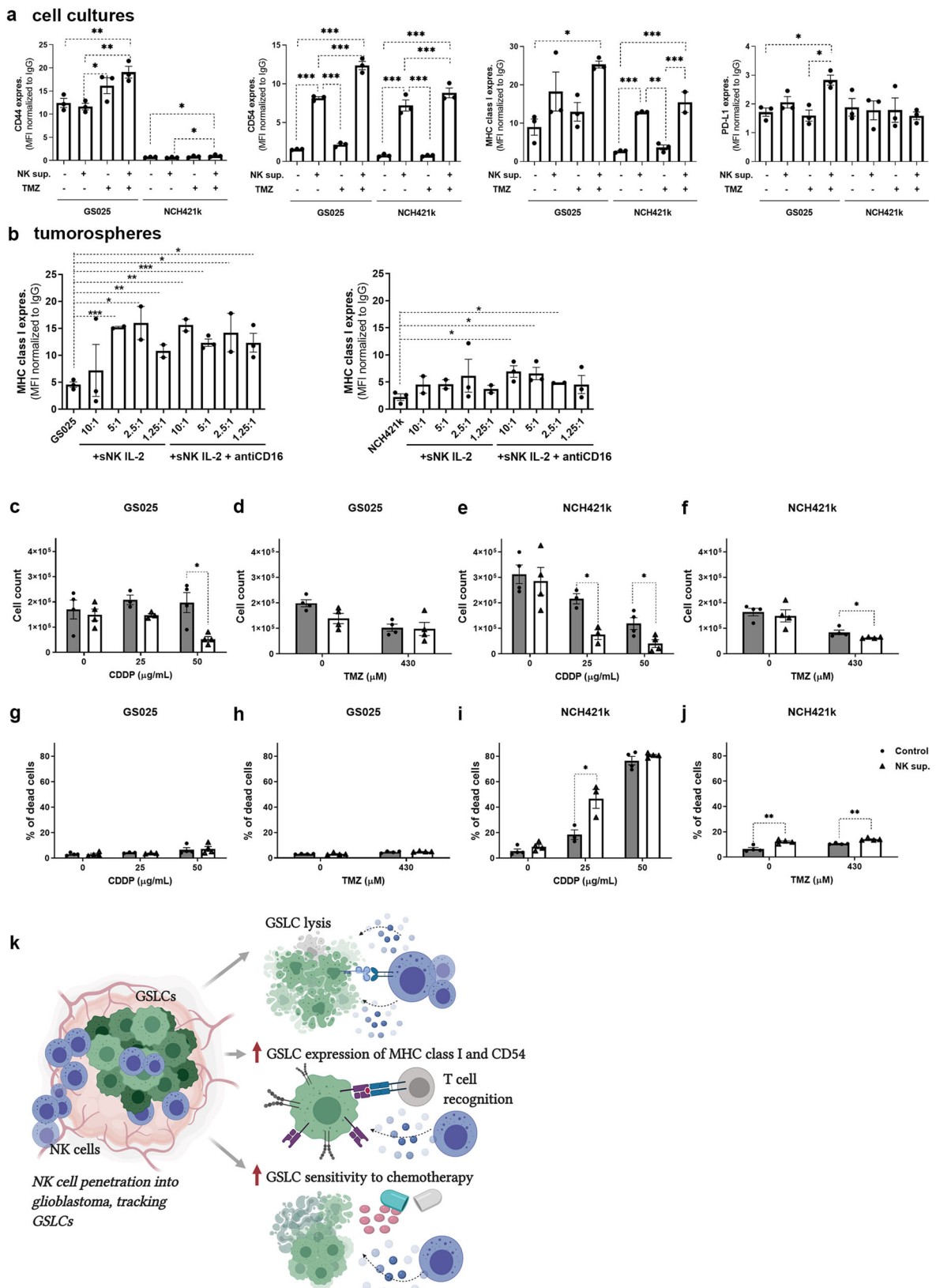

cells killed more GSLCs in both 2D and 3D in vitro models. Moreover, CD16 down-regulation in NK cells by monoclonal antibody treatments, mimicking the in vivo split anergized cells, decreased the cytotoxicity of primary and super-charged NK cells against GSLCs.

More complex 3D in vitro cellular models, including tumor-ospheres and organoids, that mimic tumor heterogeneity, cell-cell interactions, oxygen and nutrient deprivation and physical barriers in tumors in vivo[37], are needed to test immunotherapeutic strategies, such as NK cell-based approaches. In 3D GSLC models

**Fig. 6 NK cell supernatant increased cell surface expression of CD44, CD54, MHC class I and PD-L1 and cytostatic and cytotoxic effects of chemotherapeutics in GSLCs.** GSLCs were incubated with NK cell supernatants for 5 days and TMZ (430 μM) or vehicle control was added to cultures 72 h before cell surface marker expression assessment in cell cultures (n = 3, data points represent independent biological replicates) (**a**). Super-charged NK cells from at least two different heathy donors were added to tumorospheres in different NK:GSLC cell ratios for 48 h (data points represent measurement for each healthy donor). MHC class I surface expressions were evaluated in dissociated GSLCs (**b**). GS025 (**c**, **d**, **g**, **h**) and NCH421k (**e**, **f**, **i**, **j**) cells were incubated with NK cell supernatant (white bars) for 5 days. Blank NK cell culture medium without secreted factors from NK cells was used as control (gray bars). CDDP (25 and 50 μg/mL) and TMZ (430 μM) were added to cultures 24 h and 72 h, respectively, before cell count and % of dead cells assessments. Data points represent independent biological replicates (n ≥ 3). Schematic presentation of NK cell effects on GSLCs (**k**). Image was created by BioRender.com. Data are presented as means ± SEM.

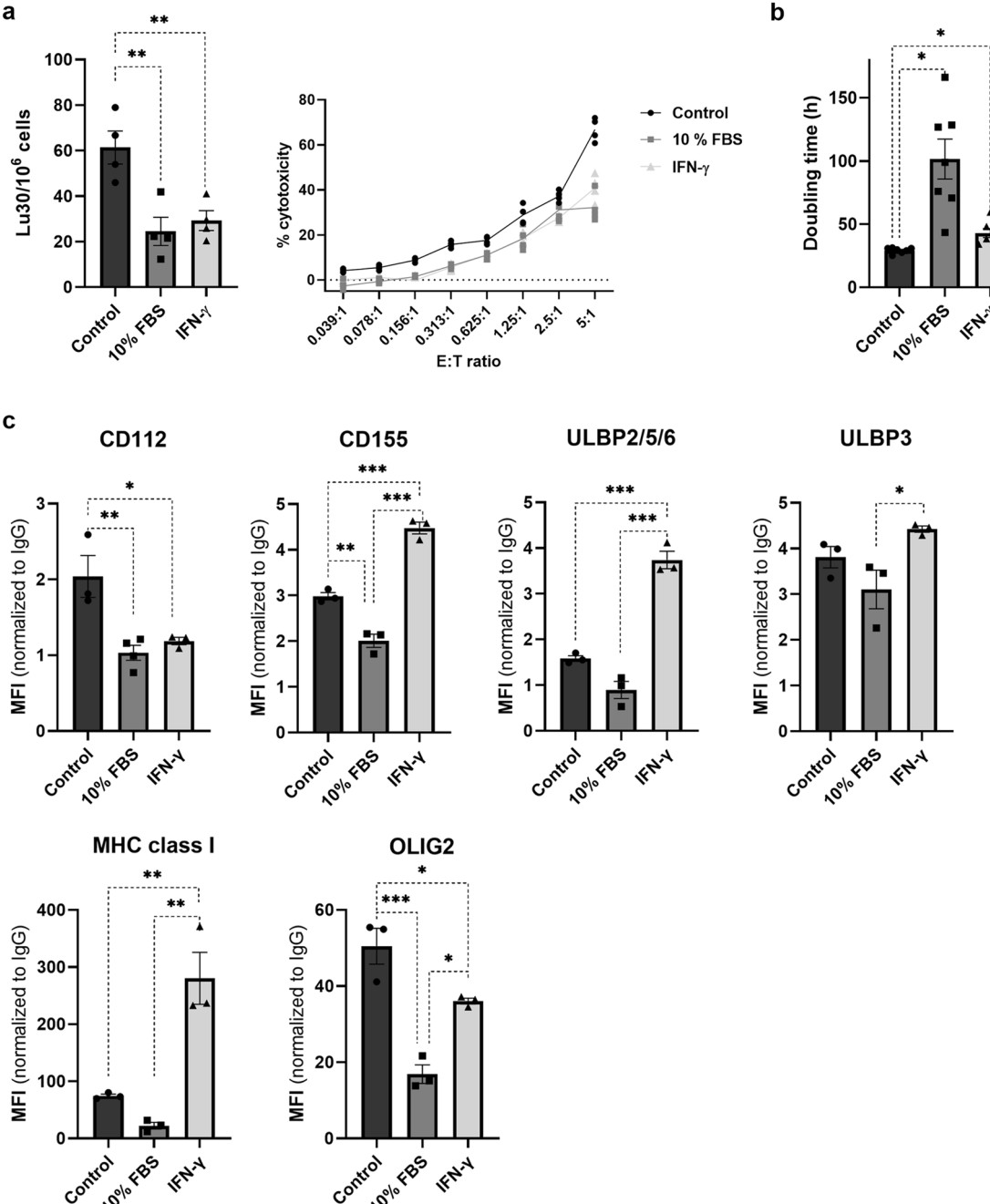

**Fig. 7 Super-charged NK cell-mediated lysis was impaired after GSLC differentiation with 10% FBS and exposure to IFN-γ. Differentiation and IFN-γ treatment decreased GSLC proliferation and altered levels of ligands for NK cell activating and inhibitory receptors as well as decreased levels of OLIG2.** After NCH421k cells were treated with cell culture medium containing 10% FBS (dark gray bars) for 7 days or 30 ng of recombinant human IFN-γ (light gray bars) for 48 h, cytotoxicity assay using calcein AM release (**a**), doubling time analysis (**b**) and protein expression analysis using immunostaining and flow cytometry (**c**) were performed. Neurobasal GSLC culture media was used as control (black bars). Data are presented as means ± SEM, data points represent independent biological replicates, n ≥ 3.

in vitro and in human GBM tumor tissue sections, we have shown that NK cells penetrate GBM tumors and can directly interact with GSLCs. NK cells penetrate the BBB, and therefore application of NK cells to GBM patients could restore NK cell function within the tumor microenvironment.

In the present study, we demonstrated that the differential susceptibility of GSLCs to NK-mediated cytotoxicity correlates with different in vitro and in vivo properties of GSLCs. Several studies have shown great GSLC plasticity suggesting that multiple cellular states of GSLCs with different differentiation/stemness programs exist in GBM tumors. They are defined by specific phenotypic properties and biomarkers[38]. As described above, we selected two models of GSLCs, GS025 and NCH421k cells, with low and high susceptibility to NK cell cytotoxicity, respectively. Slowly growing GS025 cells, more sensitive to FBS differentiation, were less susceptible to NK-mediated cytotoxicity compared to highly proliferating and tumorigenic NCH421k cells with strong stemness potential[3]. GS025 cells also expressed higher cell surface levels of CD54 and MHC I than NCH421k cells. Increased MHC I and CD54 imply decreased susceptibility of tumor cells to NK cell-mediated lysis[11]. Increased MHC I expression on tumor cell surface means that tumor cells are likely to be recognized by T cells, while decreased expression or the dysfunctional MHC class I on GSLCs may be responsible for decreased T cell recognition[24,25]. GS025, but not NCH421k cells, expressed high levels of cell surface receptor CD44. In GBMs, CD44 is characteristically overexpressed in mesenchymal GSLC subtype, associated with higher aggressiveness, therapeutic resistance and lower survival of patients with this GBM subtype[1,38]. Mesenchymal GSLC aggressiveness could also be a consequence of GSLC immune evasion from NK-mediated killing as demonstrated in our experiments with GS025 cells. All GBM cells express PD-L1, known to inhibit T cell cytotoxicity. Additional experiments with fast-growing NCH421k GSLCs showed that treatment of the cells with serum and IFN-γ induced differentiation of the cells, which expressed lower levels of the transcription factor and stem-like marker OLIG2 and proliferated more slowly, and reduced their susceptibility to super-charged NK cell-mediated cytotoxicity. This phenomenon was associated with an altered balance between ligands of activating and inhibitory NK cell receptors on NCH421k cells. Moreover, the presence of NK cells and the secretion of IFN-γ in the tumor microenvironment may trigger the differentiation of GSLCs, which are less sensitive to NK cell cytotoxicity, but express high levels of MHC class I.

NK cells do not only contribute to antitumor immunity by directly eliminating tumor cells, but also by regulating the function of other immune cells, including DCs, cytotoxic lymphocytes and macrophages, through cytokine secretion[30]. Addition of primary NK cells to GSLCs in 2D and 3D cultures increased secretion of main pro-inflammatory cytokine IFN-γ. Split anergized primary NK cells induced higher secretion of IFN-γ after interaction with GSLCs than cytotoxic primary NK cells. Higher increase in IFN-γ secretion was detected when NK cells were cultured with NCH421k cells that had higher stemness characteristics, supporting our previous studies in other types of cancer[10,11,18,23].

Elevated pro-inflammatory cytokine IL-6 has been found in GBM tumors and CSF of GBM patients[21,39]. Namely, IL-6 has been associated with pro-tumorigenic role in gliomas. IL-6 promotes GBM tumor progression in mice[39] and activates STAT3 pathway in GSLCs to maintain tumorigenic potential[40].

GBM tumors are different in that they remain persistently inflammatory even upon treatment with immune effectors, and they will increase, rather than decrease, inflammatory cytokines IL-6 and IL-8 while decreasing IFN-γ[21]. This is similar to what we see in CSF of GBM patient, since patient has increased IL-6 and

IL-8 secretion and low IFN-γ secretion. Although IFN-γ is increased in the CSF of the GBM patient with IL-2 or IL-2 and anti-CD3/CD28 mAb treatment and this results in the decreased IL-6/IFN-γ or IL-8/IFN-γ ratio, the IFN-γ secreted from these cells is likely to increase differentiation of GSLCs. However, this will result in only further decrease in IFN-γ secretion from the interacting immune effectors while the levels of IL-6 or IL-8 will continuously rise, the scenario, which we see by the interaction of NK cells with GBM cells. Thus, during the immunotherapeutic treatment of GBM patients, the levels of IL-6 or IL-8 should be targeted to minimize their effect on the growth or expansion of GBM cells[21].

Finally, studying a possible cooperation between NK cells and chemotherapeutic drugs we showed that exposure of GSLCs to NK cell supernatants increased their sensitivity to chemotherapeutic agents CDDP and TMZ. NK cell supernatants increased the cytostatic effect of CDDP in both GSLCs. However, NK cells increased TMZ cytostatic and cytotoxic effect only in fast growing and tumorigenic NCH421k tumor cells. Both chemotherapeutic drugs showed increased effect in highly tumorigenic and proliferating NCH421k after their exposure to NK supernatant, but did not show cytotoxicity in slowly proliferating GS025 GSLCs. This effect may be expected, as it is well known that DNA damaging effect of chemotherapeutics is more pronounced in faster dividing tumor cells. There is a whole spectrum of yet unknown TMZ activities that has been challenged recently[41] and as seen in present study one of them is also TMZ-mediated increase of immune-related surface markers.

We observed that NK cells can change GSLC phenotype and lower their resistance to chemotherapy. However, in this case the NK supernatants did not induce GSLC differentiation into the astrocyte-like differentiated GBM cells. Instead, NK supernatant and TMZ may have induced the GSLC trans-differentiation into another GSLC state with altered immune response to T and NK cells due to their known plasticity. NK cells and TMZ did indeed increase cell surface expression of CD44, CD54 and MHC class I, important for immune cell recognition[10,11,25]. It is possible that NK paracrine effects decreased expression of multidrug resistance genes/proteins, such as ABC (ATP-binding cassette) transporters, decreasing general GSLC resistance to chemotherapeutics[1,3,4].

In conclusions, we showed that NK cell function was impaired in peripheral blood of GBM patients in comparison to healthy donors. The presence of NK cell markers in peri-vascular GBM tissue areas indicates that NK cells crossed the leaky BBB and infiltrated the GBM tumors. Super-charged NK cells eliminated GSLCs in 2D and 3D cellular models in vitro more efficiently than the primary NK cells. Super-charged NK cell cytotoxicity was impaired after GSLC differentiation and this phenomenon was associated with altered expression levels of ligands for activating and inhibitory NK cell receptors. Besides their cytotoxicity, NK cells secreted pro-inflammatory IFN-γ and IL-6 and increased sensitivity of GSLCs to chemotherapeutic drugs, TMZ and cisplatin. Immunotherapy with allogenic super-charged NK cells appears a promising therapeutic approach in the treatment of GBM by selectively killing malignant cancer stem-like cell population, and increasing their immune-related surface markers (MHC class I and CD54) as well as their susceptibility to chemotherapy (Fig. 6k). However, known GSLC plasticity and heterogeneity is urging the research into designing a personalized approach that needs to be considered when applying NK immunotherapy in GBM treatment. To confirm our findings, in vitro and in vivo studies with higher number of patient-derived GSLCs are planned. These studies are focused on evaluating the antitumor effect of injected super-charged NK cells in GBM-bearing humanized BLT mice.

## Methods
More material details are in Supplementary file.

**Fluorescence immunohistochemistry.** Glioblastoma biopsies were obtained from glioblastoma patients who were operated at the Department of Neurosurgery, University Medical Center of Ljubljana, Slovenia. The study was approved by the National Medical Ethics Committee of the Republic of Slovenia (Approval no. 0120-190/2018/4). Altogether, 8 patients with glioblastoma (World Health Organization [WHO] glioma grade IV) were included. Tumor diagnoses were established by standard histopathology protocols at the Institute of Pathology of the Medical Faculty, University of Ljubljana. Details of GBM patients and their tumors are described in Supplementary Table 2. Formalin-fixed, paraffin-embedded tissue sections were used for immunohistochemical analyses[33,42]. Tissue sections were permeabilized and non-specific binding was blocked using a solution containing 10% fetal bovine serum (v/v), 0.1% Triton X-100 (v/v) and 1% BSA (w/v) in PBS for 1 h at room temperature. After blocking, sections were incubated with True-Black reagent (Biotium) diluted 1:20 in 70% ethanol for 30 s to block autofluorescence due to lipofuscin and blood components. Primary goat anti-human CD56, mouse anti-human CD16, mouse anti-human SOX2, rabbit anti-human SMA and rabbit anti-human CD3 were used. Alexa Fluor 488-conjugated donkey anti-goat antibodies, Alexa Fluor 546-conjugated donkey anti-rabbit antibodies and Alexa Fluor 647-conjugated anti-mouse antibodies, were used as secondary antibodies. Nuclei were stained with Hoechst 33258 solution (94403, 1:1000, Sigma-Aldrich), for 5 min at room temperature. After washing with PBS, slides were mounted in mounting solution (P36930, Invitrogen), cover-slipped, and sealed with nail polish. Confocal imaging was performed using confocal microscope (SP8 TCS, Leica) and LAS X Life Sciences software. Details of antibodies used for fluorescence immunohistochemistry are listed in Supplementary Table 3.

**Tumor cell cultures.** GSLCs NCH421k were a generous gift of prof. Christel Herold-Mende (Heidelberg University, Heidelberg, Germany)[3,42]. GS025 GSLCs were isolated from patients with GBM at University of California Los Angeles (UCLA) under UCLA Institutional Review Board (IRB) protocol 10-000655[15]. GS025 and NCH421k cells were grown in serum-free conditions as described before and were validated to express GSLC markers, such as CD133 and SOX2[3,15,33,43]. Oral squamous carcinoma stem cells (UC2) were isolated from cancer patients with tongue tumor at UCLA and cultured as described previously[21,23,26]. All cells were checked for Mycoplasma using MycoAlert Mycoplasma Detection Kit (Lonza, Switzerland). Authentication of cells was performed by DNA fingerprinting using Amp-FlSTR Profiler Plus PCR Amplification Kit, as described previously[3,44]. GS025 and NCH421k cells have a methylated MGMT promoter[15,45].

**Purification and treatment of primary NK cells.** NK cells were purified from healthy donors with no clinical signs of disease and patients diagnosed with GBM using EasySep Human NK Enrichment kit (STEMCELL technologies, Canada) as described previously[19,21,23]. Freshly purified primary NK cells were then activated with IL-2 or combination of IL-2 (1000 units/mL) and anti-CD16 monoclonal antibody (mAb; 3 μg/mL) for 18–24 h to obtain primary cytotoxic and split anergized NK cells, respectively. In addition, PBMCs and NK cells were purified from three patients diagnosed with GBM. GBM patient PBMCs and NK cells were also treated with combination of IL-2 (1000 units/mL) and anti-CD3/28 mAb (25 μL/mL) or sonicated pro-biotic bacteria AJ2[19,23] for 18–24 h to activate T cells and induce IFN-γ secretion, respectively. The studies with immune cells from peripheral blood of healthy donors and GBM patients were approved by the UCLA Institutional Review Board (IRB#11-000781), and all participants signed written informed consent in accordance with the Declaration of Helsinki.

**Generation of osteoclasts and expansion of super-charged NK cells.** Generation of osteoclasts and super-charged NK cells from heathy donors was performed as described previously[19,23]. Briefly, purified human monocytes were differentiated to osteoclasts by treatment with M-CSF (25 ng/ml) and RANKL (25 ng/ml) for 21 days. Human purified primary NK cells were activated with IL-2 (1000 units/ml) and anti-CD16 mAb (3 μg/ml) for 18–20 h before they were cocultured with osteoclasts as feeder cells and sonicated pro-biotic bacteria AJ2[19,23]. The culture media was refreshed with IL-2 every 3 days. The number of NK cells and contamination with T cells was monitored during super-charged NK expansion by flow cytometric analysis of CD16/56 and CD3 immunostaining. Not older than 1 month expanded super-charged NK cells were used for experiments. Details of antibodies used for flow cytometry are listed in Supplementary Table 4.

**2D cocultures of NK cells and GBM cells.** NK cells were left untreated or treated with IL-2 or IL-2 and anti-CD16 mAb for 18–24 h, before they were cocultured with GBM cells (single cell suspension) in different ratios for 24 h.

**3D GBM model- tumorospheres.** GBM cells were labelled with 30 μM of Cell-Tracker Green for 30 min. Then GBM cell tumorospheres were established as described previously[3,44] and incubated for 6 days in cell incubator (37 °C, 5% $CO_2$).

NK cells were labelled with 10 μM CellTracker Blue for 30 min before they were added to GBM cell tumorospheres in different ratios in U-bottom 96-well plate for 4–48 h. Tumorospheres were dissociated using a mixture of TrypLE Express and Collagenase type II for 30 min and cells were stained to assess cell count with Trypan blue staining, cell death using propidium iodide (PI) and flow cytometry and marker surface expression using immunolabelling and flow cytometry. Number of dead GSLCs was determined as % of CellTracker green-gated dead cells (PI-positive) and multiplied by the number of GSLCs in control spheroids.

**Flow cytometry.** Surface staining was performed as described previously[23,46]. PBMCs from patients and heathy donors were stained with antibodies against CD45, CD3, CD4. CD8, CD14, CD19, CD56/CD16 and analyzed with flow cytometry to determine % of immune cell populations within the CD45-positive cell population. Tumor cells were labelled with PE-conjugated antibodies against CD44, CD54, MHC class I, PD-L1 and B7-H6, FITC-conjugated antibodies against CD155 and ULBP1, unconjugated antibodies against ULBP3 and APC-conjugated antibodies against MICA/B, ULBP2/5/6, CD112, and HLA-E, and incubated for 30 min at 4 °C. After washing samples were analyzed by Attune flow cytometer (Invitrogen) and FlowJo software (Ashland, OR, USA). IgG isotypic controls were used as controls. PI in final concentration of 10 μg/mL was added to cell suspension to evaluate cell death. GFAP, OLIG2, and CD133 intracellular staining was performed as described previously[3] and measured by flow cytometer. Primary monoclonal mouse anti-human GFAP antibody, monoclonal rabbit anti-human OLIG2 antibody, polyclonal rabbit anti-human CD133 antibody were used. Secondary PE-conjugated anti-mouse secondary antibody and Alexa Fluor 488-conjugated goat anti-rabbit antibody were used for GFAP and OLIG2/ CD133 staining, respectively. Details of antibodies used for flow cytometry are listed in Supplementary Table 4.

**Cytotoxicity assay - Chromium release assay.** The $^{51}$Cr release assay was performed as described previously[18,19,21,26]. Briefly, different numbers of purified NK cells were incubated with $^{51}$Cr–labeled tumor target cells. After a 4 h incubation, the supernatants were harvested from each sample and counted for released radioactivity using the gamma counter. The percentage of specific cytotoxicity was calculated as follows (Equation 1):

$$\%\text{Cytotoxicity} = \frac{(\text{Experimental cpm} - \text{Spontaneous cpm})}{(\text{Total cpm} - \text{Spontaneous cpm})}$$

LU (lytic unit) 30/10$^6$ is calculated by using the inverse of the number of effector cells needed to lyse 30% of tumor target cells × 100.

Cytotoxic ability of patient-derived primary NK cells was always tested on UC2 cells.

**ELISA.** ELISA kits for human IFN-γ and IL-6 (Biolegend, San Diego, CA, USA) were used according to manufacturer's instructions.

**ELISPOT.** The number of IFN-γ secreting cells was analyzed using Human IFN-γ Single-Color Enzymatic ELISPOT Assay, ImmunoSpot® S6 UNIVERSAL analyzer and ImmunoSpot® SOFTWARE (all CTL Europe GmbH, Bohn, Germany), according to manufacturer's instructions.

**Multiplex cytokine assay.** Multiplex arrays were used to determine the levels of secreted cytokines and chemokines. Analysis was performed using MILLIPLEX MAP Human High Sensitivity T Cell Panel (Millipore, Danvers, MA, USA) and data was analyzed using xPONENT 4.2 software (Luminex, Austin, TX, USA).

**Proliferation analysis in vitro.** GBM cells ($5 \times 10^4$) were seeded in 12-well plate and cultured for 72, 96 and 120 h. Cells were harvested and doubling time was calculated as follows (Equation 2):

$$Doubling\ time(h) = \frac{incubation(h) \times \log(2)}{\log(final\ conc.) - \log(initial\ conc.)}$$

Doubling time in GBM tumorospheres was calculated based on cell numbers after dissociation by the same formula.

**Confocal imaging and analysis of tumorospheres.** GBM tumorosheres in U-bottom well plate were imaged from 4 h to 48 h after addition of NK cells under ×100 magnification using inverted laser scanning confocal microscope with Air-yScan LSM880 and ZEN black software (both Zeiss, Germany). CellTracker blue was excited with laser line at 405 nm and emission was detected between 410 and 485 nm. CellTracker green and PI were excited with laser line at 488 nm and emission was detected between 495 and 550 nm for CellTracker green and 570–700 nm for PI. Laser power, gain and offset were kept constant between experiments and conditions. Z-stacks of confocal sections were acquired with a step size of 2.99 μm. Z-stack images of tumorospheres were analyzed and reconstructed using ZEN blue software (Zeiss, Germany) and Imaris (Bitplane) software version 9.5.1 (Oxford Instruments, United Kingdom). The number of GBM cells and dead cells was quantitated with the spot detection tool of the Imaris software. For

analysis of direct cellular interactions, surface 3D renderings were created using surface area module for NK cell (CellTracker blue) and GSLC (CellTracker Green) surface using Imaris software. We then obtained surface reconstructions of NK cell surface touching GSLC surface in gray color using surface-surface contact area plug in.

**Treatment of GSLCs with NK supernatants and IFN-γ.** GSLCs were exposed to a total of 30 ng of human recombinant IFN-γ for 48 h and analyzed for surface marker expression. Treatment of GSLCs with NK supernatants was conducted as described previously[18]. Briefly, activation of NK cells was done by treatment with a combination of IL-2 (1000 U/mL) and anti-CD16 mAb (3 µg/mL) for 18 h before the supernatants were removed and used for further experiment. The amounts of IFN-γ produced by activated NK cells were assessed using ELISA kit as described above. To treat GSLCs a total of 3900 pg of IFN-γ containing supernatants were added for 5 days. After that these GSLC cells were treated with cytotoxic drugs cisplatin (CDDP, cis-diamminedichloridoplatinum(II), in final concentrations of 25 µg/mL and 50 µg/mL) and temozolomide (TMZ in final concentration of 430 µM) for 24 h and 72 h, respectively. Cell culture media and drug vehicle DMSO were used as control for CDDP and TMZ treatment, respectively. Cell number and cell death was evaluated using Trypan blue and PI staining.

**Cytotoxicity assay – Calcein AM release assay.** After NCH421k cells were grown in Neurobasal culture medium (Gibco, ThermoFisher Scientific) supplemented with 10% FBS for 7 days or exposed to 30 ng of human recombinant IFN-γ for 48 h, they were tested for super-charged NK cell-mediated cytotoxicity. Super-charged NK cells were activated overnight with 1000 IU IL-2/mL. Super-charged NK cells were prepared in selected effector to target ratios. NCH421k target cells were labeled with 15 µM calcein-AM stain in serum-free media for 30 min. After a washing step, 5000 target cells were added to effector cells. The plate was centrifuged at 200 g for 1 min and incubated for 3 h at 37 °C with 5% CO2. After incubation, the plate was centrifuged at 700 g for 5 min, and 50 µl supernatant was transferred to a new microtiter plate for fluorescence measurements. Released calcein-AM was measured using microplate reader Infinite M1000 Tecan (Tecan, Switzerland) at 496 nm excitation and 516 nm emission. The percentage of cytotoxicity was calculated as: 100 × (test release−spontaneous release)/(total release −spontaneous release). Spontaneous release of calcein-AM was measured in wells containing 100 µl super-charged NK culture medium and 50 µl target cells. For total release, 2% Triton X-100 was added to the super-charged NK culture media to achieve target cell lysis. Lytic units (LU) were calculated using the inverse of the number of effector cells needed to lyse 30% of the target cells multiplied by 100.

**Intracranial mice injections.** Animal research was performed at UCLA under the written approval of the UCLA Animal Research Committee (ARC) (protocol # 2012-101-13). Analysis of GBM cell growth was analyzed in immune-deficient NSG mice (Jackson Laboratory, USA) after intracranial injection of GS025 and NCH421k cells. Briefly, GBM cells were injected (4 × 10^5 cells per injection) into the right striatum of the brain in female NSG mice (3 weeks old). Injection coordinates were 2 mm lateral and 1 mm posterior to the bregma, at a depth of 2 mm[15]. Mice were sacrificed when clinical signs of brain tumor developed.

**Tumor burden measurements in mice and isolation of tumor cells from mice tumors.** Tumor burden was monitored on the basis of secreted *Gaussia* luciferase measurements in mice blood. GBM cells were infected with a lentiviral vector containing a secreted *Gaussia* luciferase (sGluc)-encoding reporter gene (Targeting Systems no. GL-GFP) and intracranially implanted into the right striatum in mice. Measurements of sGluc levels were performed as described previously[15]. GBM mice tumors were minced, washed with 1× PBS, incubated with trypsin for 10 min and passed through 70 µm strainer. Isolated single human GSLC (GS025, NCH412k) suspensions were incubated in serum-free conditions and imaged for several days.

**Statistics and reproducibility.** All experiments were set up in duplicates or triplicates and tested in at least three independent experiments ($n \geq 3$) unless stated otherwise. An unpaired, two-tailed Student's $t$ test or one-way ANOVA test with Bonferroni correction was used to perform statistical analyses in GraphPad Prism software version 8 (La Jolla, CA, USA). $P$ values < 0.05 were considered to indicate significant differences. The $p$ values were expressed within the Figures as follows: \*\*\*$p$ value < 0.001, \*\*$p$ value: 0.001–0.01, \*$p$ value: 0.01–0.05.

**Reporting summary.** Further information on research design is available in the Nature Research Reporting Summary linked to this article.

## Data availability

All data generated or analyzed during this study are included in this published paper and its Supplementary File and Supplementary Data File 1. The datasets generated during and/or analyzed during the current study are also available from the corresponding author on reasonable request.

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

## Acknowledgements

We acknowledge the contribution and support of prof. Tamara Lah Turnšek from National institute of biology Ljubljana. We thank Andrej Porčnik, MD, from University Medical Centre Ljubljana for clinical data of glioblastoma patients. This work was supported by Slovenian Research Agency (Grant programs P1-0245 and P1-0207, Grant project J3-8201, Postdoctoral project Z3-1870, Young researcher grant and Bilateral project BI-US/19-21-021) and by the European Program of Cross-Border Cooperation for Slovenia-Italy Interreg TRANS-GLIOMA. We thank our funding agencies (NIH and NIDCR) and all the donors for supporting the work.

## Author contributions

B.B. designed and performed experiments, analyzed and interpreted data and wrote the paper. M.W.K. designed and performed experiments and analyzed data. C.T., P.C.C., E.S., B.M., A.H., N.A. performed experiments and analyzed data. M.N., V.Ž., J.M. provided samples and analyzed the data. D.N. analyzed and interpreted data. A.J. conceived the project, designed the experiments, analyzed and interpreted the data and wrote the paper. All authors reviewed and approved the submitted version of paper.

## Competing interests

The authors declare no competing interests.
