## [Peer Review File · Communications Biology]

Reviewers' comments:

Reviewer #1 (Remarks to the Author):

The study from Breznik and colleagues is very interesting. The experimental design appears very developed and the manuscript is well written.

NK cells are an innate immune component with a strong potential in tumor cell elimination. They have an important "liaison" with DC and collaborate with CD4+ T cells in taking under control tumor progression.

Based on their critical role as effector cells, the authors should stress the description of NK cell implication in antitumor cancer immunity. There are some clinical studies describing the critical implication of NK cells, that should be addressed.

I also suggest verifying whether infiltrating CD56 cells also express CD8.

IL-15 is not investigated or considered. Please address the potential role of this important cytokine.

It would be useful to support the conclusions of the study with an in vivo experiment (survival analysis).

Reviewer #4 (Remarks to the Author):

Infiltrating NK cells bind, lyse and increase chemotherapy efficacy in glioblastoma stem-like tumorspheres, by Breznik et al.,

In this work, Breznik and colleagues investigated the biological activity of so-called super-charged NK cells (split-anergy phenotype produced using IL-2+ antiCD16 on osteosarcoma cells with sonicated probiotic bacteria AJ2) against glioma stem-like cells, (GSLCs) by probing NK cell-tumor interactions, cytotoxicity, cytokine secretion and tumor penetration in vivo in animal models. They used various methods, two GSLCs cell line models, n=8 GBM biopsy derived tissue, patient derived cerebrospinal fluid (CSF) as well as PBMCs from healthy donors as comparator.

This is an interesting work where the authors have placed a substantial experimental effort in developing their story. Nevertheless, the work lacks mechanistic insights to elucidate precisely how or why the super charged NK cells preferentially kill the fast growing GSLCs but fail to control the slower growing cells. How does TMZ upregulate MHC -I, CD54 expression in GSCLs treated with NK supernatant as opposed to NK supernatants alone without TMZ.. what is the mechanism of effect? Sensitization of NK cell supernatant to TMZ induced GBM Cell death in NCH421K cells in this short term assay (Fig 6j) is very marginal.. looks like just a marginal NK effect to this reviewer... what is the MGMT promoter methylation status of this cell line? This is relevant since TMZ 430µM does not substantially increase NCH421K cells in death compared no TMZ. The authors use CD56 staining I biopsy specimens to visualize NK cells in vascular niches in close proximity with sox2 positive immature cells. They claim that CD44 is a putative GSLC marker although this is controversial when used in context of immunohistochemistry since levels of expression cannot be determined by this method (PMID: 29218080), only by flow cytometry. As well as NK cells, T cell and GBM cells also express CD56 (Todaro et al., 2007), therefore in addition to showing the that CD56+ cells did not express GFAP, or CD3.. the authors could have also added an NK marker such as NKp46. This analysis is purely phenomenological and is it not quantified and not clear how many biopsies the data that are presented from. The legend states n=4 but appears to refer to the PBMCs data and then it is not clear if this was also the case for the immunohistochemistry analyses. How and why were only these 4 /8 selected? It is not stated what target cells were used in data presented in Fig 1j and Fig1g. Are these data from both GSCLCs or just one of these? It is not clearly specified in the legend. Regarding data in Fig 1f, are these numbers from GBM peripheral blood or is this tumor derived? If indeed this is tumor derived, the number of NK cells within the CD45+ gate are much higher than previously reported in the literature, then it is imperative to establish that the

cell harvesting procedure was not contaminated by blood product. As NK cell repopulation is variable, it would be informative to have data from several targets. It is therefore a weakness of the paper that they have conducted these experiments in only 2 cell lines. The studies with CSF are not appropriately described in the methods section (ethics, experimental details e.g. volume normalization, site of tapping etc) and results. It is not mentioned how CSF was tapped, whether it was through a sterile lumbar puncture procedure or whether this fluid was obtained from an intraoperative procedure. The latter can conceivably be associated with blood contamination, a source of error that may affect the high numbers and type of immune cells detected in this CSF. More information is required. Introductory text to NK biology and manuscript as a whole can benefit from greater clarity.

ANAHID JEWETT, Ph.D., M.P.H., PROFESSOR
DIVISION OF ORAL BIOLOGY & MEDICINE
BOX 951668, B389 088
LOS ANGELES, CA 90095-1668

SCHOOL OF DENTISTRY
CENTER FOR THE HEALTH SCIENCES
10833 LE CONTE AVENUE, BOX 951668
LOS ANGELES, CALIFORNIA 90095-1668

Point-by-point response to the referees' comments

Dear Editor and Reviewers,

We are pleased that we have been given the opportunity to revise our manuscript. We thank you for all valuable and insightful comments that enable us to significantly improve the manuscript. In this letter, the authors of the manuscript respond to the two reviewer reports on the basis of which the editorial advice is "Major revision". All amendments in the manuscript are made as track changes.

Reviewer #1

Comment: The study from Breznik and colleagues is very interesting. The experimental design appears very developed and the manuscript is well written.

NK cells are an innate immune component with a strong potential in tumor cell elimination. They have an important "liaison" with DC and collaborate with CD4+ T cells in taking under control tumor progression.

Reply: *We thank the reviewer for positive feedback and comments. We agree that besides other functions (e.g. effector) NK cells are important as regulatory cells and in collaboration with other immune cells, such as dendritic cells and CD4+ T helper cells, control tumor progression (Vivier 2008; Barry 2018). In line with this, we added the description of this important role in the Introduction. Please see lines 104-108: "In addition to cytotoxic function, NK cells act as regulatory cells and also secrete various pro- and anti-inflammatory cytokines and chemokines, such as IFN- γ and interleukin (IL)-6, which that orchestrate innate and adaptive immune responses. For example, NK cells boost the tumor infiltration as well as maturation and activation of DCs and T cells and by that promote anti-tumor immune responses."*

Comment: Based on their critical role as effector cells, the authors should stress the description of NK cell implication in antitumor cancer immunity. There are some clinical studies describing the critical implication of NK cells, that should be addressed.

Reply: *NK cells and NK cell-directed immunotherapies are currently in several preclinical and clinical studies for treatment of several hematological and solid malignancies due to their role in anti-tumor immune response – effector functions and regulatory function of adaptive and innate immune system (Myers and Miller 2021; Biederstadt and Rezvani 2021). Several studies have shown encouraging clinical responses of NK cell-based therapy and to be safe. We included sentences on implication of NK cells in antitumor immunity and in clinical studies to the Introduction, please see lines 116-123:*

"Due to their crucial role in antitumor immune responses, NK cells and NK cell-directed immunotherapies are currently in several preclinical and clinical studies for treatment of hematological and solid malignancies. These studies have shown encouraging clinical responses and have been shown to be safe in cancer patients. Despite the great potential of NK cell-based therapies, there are still many challenges to translate the use of NK cells into the clinical practice. These include, including prolonging the persistence of NK cells in vivo persistence and overcoming their exhaustion, as well as in vitro expansion to obtain

sufficient quantity of efficient therapeutic NK cells.”

Comment: I also suggest verifying whether infiltrating CD56 cells also express CD8.

Reply: *We thank the reviewer for this excellent suggestion. We have performed immunofluorescence staining of CD8 in GBM tissue sections, together with NK cell markers CD56 and NCR1 (NKp46). Tumor-infiltrating NK cells do not express CD8. The results are commented in the manuscript (lines 158-160) and are shown in Supplements as Supplementary Figure 4.*

Comment: IL-15 is not investigated or considered. Please address the potential role of this important cytokine.

Reply: *We are aware that IL-15 is important cytokine for NK cell activation and expansion. In our study, IL-2 was used to activate primary NK cells in vitro. This is based on all our previous studies that have shown cytotoxic activation of primary NK cells against tumor cells of several types of cancer, including glioblastoma (Tseng 2015; Kozłowska 2016; Kaur 2017; Kaur 2019; Kaur 2021). IL-15 and IL-12 are also important cytokines for NK cell activation and expansion and are produced and secreted from osteoclasts that were used in our study as feeder layer for establishment of super-charged NK cells (Tseng 2015). Super-charged NK cells also secrete high amounts of IL-15 when compared to IL-2 activated primary NK cells as has been already proved by our laboratory. Furthermore, if IL-15 signaling was blocked in culture of super-charged NK cells using monoclonal antibodies, the cytotoxic function and NK cell expansion was reduced. In contrast, blocking of IL-12 in super-charged NK cells resulted in decreased secretion of IFN- γ (Kaur 2017).*

We addressed the IL-15 role in NK cell activation and expansion in the Introduction (lines 113 – 114). Furthermore, we added the importance of IL-15 and IL-12 for NK cell activation and expansion in the Discussion. Please see lines 99-100, 128-130 and 363-367:

“Various cytokines, including IL-2, IL-15 and IL-12, can be used to activate and expand NK cells in vitro.”

“Super-charged NK cells are produced using sonicated probiotic bacteria AJ2 and osteoclasts as feeder cells that provide all necessary signals to activate and expand NK cells, including cytokines IL-12 and IL-1⁰.”

“One example of signals that are provided by osteoclasts are IL-15 and IL-12. Namely, IL-15 blocking resulted in impaired cytotoxicity and expansion of super-charged NK cells. NK-cell mediated cytotoxicity and IFN- γ secretion were decreased in super-charged NK cell cultures after IL-12 blocking.”

Comment: It would be useful to support the conclusions of the study with an in vivo experiment (survival analysis).

Reply: *We agree with the reviewer and this is our next step to test the effect of super-charged NK cells against glioblastoma in humanized BLT mice. In line with this, we plan to analyze tumor size, NK cell infiltration in to the tumor and survival of GBM-bearing mice after intravenous injection of super-charged NK cells. We added a sentence regarding animal experiments in discussion part (please see lines 476-478):*
“To confirm our findings, in vitro and in vivo studies with higher number of patient-derived GSLCs are planned. These studies are focused on evaluating the antitumor effect of injected super-charged NK cells in GBM-bearing humanized BLT mice.”

Reviewer #4

Infiltrating NK cells bind, lyse and increase chemotherapy efficacy in glioblastoma stem-like tumorspheres, by Breznik et al. In this work, Breznik and colleagues investigated the biological activity of so-called super-charged NK cells (split-energy phenotype produced using IL-2+ antiCD16 on osteosarcoma cells with sonicated probiotic bacteria AJ2) against glioma stem-like cells, (GSLCs) by probing NK cell-tumor interactions, cytotoxicity, cytokine secretion and tumor penetration in vivo in animal models. They used various methods, two GSLCs cell line models, n=8 GBM biopsy derived tissue, patient derived cerebrospinal fluid (CSF) as well as PBMCs from healthy donors as comparator.

Comment: This is an interesting work where the authors have placed a substantial experimental effort in developing their story. Nevertheless, the work lacks mechanistic insights to elucidate precisely how or why the super charged NK cells preferentially kill the fast growing GSLCs but fail to control the slower growing cells.

Reply: *We thank the reviewer for pointing the fact that the study lacks mechanistic insights why or how super-charged NK cells are lysing GSLCs that are fast growing cells.*

In this study, we have shown that super-charged NK cells are significantly more cytotoxic than primary NK cells towards both, slowly and fast growing GBM stem-like cells (GSLCs) (Figures 1c and 4c). To reveal more insights why super-charged NK cells preferentially kill fast growing and high tumorigenic NCH421k GSLCs we performed additional experiments with NCH421k GSLCs that were exposed to serum differentiation (10 % fetal bovine serum (FBS) for 7 days) and IFN- γ (30 ng for 48h). Cell culture medium with 10% FBS has been proved to trigger GSLC differentiation (Podergajs 2016). IFN- γ , secreted from activated immune cells, has been already shown to decrease cancer cell susceptibility to NK cell lysis (Kozłowska 2016). After NCH421k GSLC treatment cells were analyzed for susceptibility to super-charged NK cell-mediated lysis, proliferation rate, protein expression of GBM and GSLC markers as well as cell surface expression of ligands of activating and inhibitory NK cell receptors and surface molecules that has been shown in the past to affect NK cell-mediated lysis.

We performed additional experiments and have shown that treatment of NCH421 cells with 10% FBS and IFN- γ led to 65 and 60% decrease in super-charged NK cell cytotoxicity, respectively. 10% FBS- and IFN- γ -treated NCH421k cells are less proliferative and express lower protein levels of GSLC marker OLIG2. Decreased super-charged NK cell cytotoxicity in 10% FBS-treated cells was associated with decreased levels of activating NK cell ligands (CD112, CD155) as compared to NCH421k control. In IFN- γ -treated cells, decreased super-charged NK cell cytotoxicity was associated with decreased level of activating NK cell ligand CD112, increased levels of activating NK cell ligands CD155 and ULBP2/5/6 as well as increased level of inhibitory NK cell ligand MHC class I as compared to NCH421k control. There were low or no expression of other ligands of activating (B7-H6, ULBP1, MICA/B) and inhibitory (HLA-E, PD-L1, CD54) NK cell receptors in NCH421k cells. Expression of astrocytic marker GFAP and GSLC marker CD133 was not altered in NCH421k cells after treatment. Based on these findings, we can conclude that super-charged NK cells preferentially lyse fast proliferating stem-like GBM cells with high OLIG2 expression. Serum differentiation and IFN- γ treatment of GSLCs impaired super-charged NK cell-mediated lysis and was associated with altered levels of several ligands for NK cell activating (CD112, CD155, ULBP2/5/6, ULBP3) and inhibitory (MHC class I) receptors. Please see lines 304-326, 406-414, 466-468 and new Figure 7. All additional experiments are described under Methods (cytotoxicity assay –Calcium AM release assay and flow cytometry analyses) .

Based on our previous studies with super-charged NK cells, we have shown that super-charged NK cells, expanded and activated using osteoclasts as feeder layer, are highly cytotoxic cells (Kaur 2017; Kaur 2019; Kaur 2020) and express high levels of activating and inhibitory NK cell receptors and activation markers, including NKG2D, NKp44, NKp46, CD54, KIR2, CD56, ki-67 and granzymes A and B (manuscript in revision). This implies that super-charged NK cells have high granule content with potent cytotoxic function

through granule-mediated lysis of GSLCs. Moreover, osteoclast-expanded NK cells showed high induction of Fas ligand and TNF- α in our previous studies (Kaur 2020). Therefore, it is also possible that the lysis of GSLCs is mediated through death receptors in GSLCs.

Comment: How does TMZ upregulate MHC -I, CD54 expression in GSCLs treated with NK supernatant as opposed to NK supernatants alone without TMZ.. what is the mechanism of effect?

Reply: As the reviewer mentioned we observed increased in cell surface expression of MHC class I and CD54 after treatment of cells with temozolomide (TMZ) and NK cell supernatant together, compared to TMZ treatment alone. We did not explore the mechanisms of this combined effect of TMZ and NK cell supernatants and it is one of the crucial points of our further investigations. The effects of TMZ on immunological characteristic of glioma stem-like cells (GSLCs) has been already confirmed by some studies (Fritzell 2013; Zhang 2017). Some of these studies have shown that MHC class I expression is increased in TMZ-treated GSLCs by NF- κ B signaling. As we observed that NK supernatant alone and NK-secreted IFN- γ can increase the expression of MHC I, we can imply on synergistic effect of NK cell supernatants and TMZ. Additional studies have to be performed to confirm this hypothesis. Cell surface expression of CD54 was already increased if GSLCs were exposed to NK cell supernatant, what is the mechanism of action with significant increases of CD54 after addition of TMZ to NK cell supernatant-exposed cells, but it remains to be elucidated in our future studies.

Comment: Sensitization of NK cell supernatant to TMZ induced GBM Cell death in NCH421K cells in this short term assay (Fig 6j) is very marginal.. looks like just a marginal NK effect to this reviewer... what is the MGMT promoter methylation status of this cell line? This is relevant since TMZ 430 μ M does not substantially increase NCH421K cells in death compared no TMZ.

Reply: It is known that both GSLCs, NCH421k and GS025, have methylated MGMT promotor. This information has been added to Methods under tumor cell cultures. Please see the lines 517-518:

“GS025 and NCH421k cells have a methylated MGMT promotor.”

TMZ alone had effect on both GSLCs, it decreased cell number of GSLCs (Figures 6d, f) and increased the number of dead cells (Figures 6h, j). The additional sentence was added to the Results, describing the increase of % of death cells after TMZ and NK supernatant treatment of NCH421k cells. Please see lines 299-300:

»However, NK cell supernatants alone, without addition of TMZ, already increased cell death of NCH421k cells.«

Comment: The authors use CD56 staining I biopsy specimens to visualize NK cells in vascular niches in close proximity with sox2 positive immature cells. They claim that CD44 is a putative GSLC marker although this is controversial when used in context of immunohistochemistry since levels of expression cannot be determined by this method (PMID: 29218080), only by flow cytometry. As well as NK cells, T cell and GBM cells also express CD56 (Todaro et al., 2007), therefore in addition to showing the that CD56+ cells did not express GFAP, or CD3. The authors could have also added an NK marker such as NKp46.

Reply: We thank the reviewer for pointing out the issue with CD44 immunofluorescence detection on paraffin tumor tissue sections. Because of this, we used several GBM and GSC markers, such as SOX2, GFAP and CD44, and immune cell markers CD45, CD3, and additionally CD8 to discriminate between GBM cells/GSLCs and immune cells/NK cells. NCR1 (NKp46) protein expression was tested in 6 GBM paraffin sections using immunofluorescence and selective antibodies for human CD56, CD8 and NCR1. NCR1 protein expression was detected only in 1 GBM patient tissue section, partly colocalizing with NK cell marker CD56 in 1 GBM tissue section. Please see Supplementary Figure 4. This is in line with recent studies that showed NK cell infiltration into GBM tumors but these tumor infiltrating NK cells displayed altered phenotype and function from peripheral blood NK cells from the same patients. Tumor infiltrating NK cells expressed higher levels of inhibitory NK cell receptors and lower levels of activating NK cell

receptors, including NCRI (Nkp46), as compared to peripheral blood NK cells from GBM patients and healthy controls (Shaim 2021).

Comment: This analysis is purely phenomenological and is it not quantified and not clear how many biopsies the data that are presented from. The legend states n=4 but appears to refer to the PBMCs data and then it is not clear if this was also the case for the immunohistochemistry analyses. How and why were only these 4 /8 selected?

Reply: Here, we wanted to show that NK cells infiltrate GBM tumors and they localize in close proximity of tumor vessels (peri-vascular regions), by immunofluorescence in GBM paraffin tissue sections. We apologize for not being clear on the following details: Immunofluorescence on GBM tissue sections was performed in 8 GBM patients. For 6 GBM tissue images are shown in Figure 1a and Supplementary Figures 1-4. The 2 GBM tissues did not contain CD56 positive cells and were not presented here. On the other hand, peripheral blood immune cell number and functions are shown in Figure 1f-l and were assessed in 4 additional GBM patients. These 4 patients were not the same patients, of which immunofluorescence in GBM tissue sections was performed. The number of GBM patients and healthy donors that were involved in assessing peripheral blood immune cell number (4 GBM and 5 healthy donors) and cytotoxicity functional assays (3 GBM and 3 healthy donors) was added in the Results (lines 164-166) and Methods (lines 528-529). There are difficulties to obtain sufficient number of patient- peripheral blood PBMCs and primary NK cells for cytotoxicity and IFN- γ secretion analyses. This was the case in 1 GBM patients and thus immune cells isolated from 3 GBM patients were tested for functional assays.

Comment: It is not stated what target cells were used in data presented in Fig 1j and Fig 1g. Are these data from both GSCLCs or just one of these? It is not clearly specified in the legend.

Reply: Oral squamous carcinoma stem cells (UC2) target cells were used to determine cytotoxicity of NK cells from GBM patients. This information is written under Methods (line 586). We added this information in legend to Figure 1. Cytotoxicity assay using chromium release and UC2 as target cells is optimized and routinely performed in our laboratory to assess immune cell number and their functions in healthy donors and cancer patients (Kaur 2017; Kaur 2020; Kaur 2021; Tseng 2010). In Figure 1a, it can be seen that susceptibility of NCH421k and UC2 cells to primary NK cell-mediated cytotoxicity was comparable.

Comment: Regarding data in Fig 1f, are these numbers from GBM peripheral blood or is this tumor derived? If indeed this is tumor derived, the number of NK cells within the CD45+ gate are much higher than previously reported in the literature, then it is imperative to establish that the cell harvesting procedure was not contaminated by blood product.

Reply: We apologize for confusion. Immune cell numbers in Figure 1g are from peripheral blood of GBM patients and healthy donors. Source of CD45-positive population was added in legend to Figure 1 (lines 829-830):

»Assessment of immune cell percent within CD45-positive population in peripheral blood was determined by immunolabeling and flow cytometry.«

Comment: As NK cell repertoires are variable, it would be informative to have data from several targets. It is therefore a weakness of the paper that they have conducted these experiments in only 2 cell lines.

Reply: We agree with reviewer that NK cell responses are variable. In this study, we want to study the effect of super-charged NK cells on GSCLCs with different phenotype, e.g., susceptibility to NK cell-mediated lysis, proliferation and stem-like properties. We are aware that further studies need to be performed with more GBM cell lines and in mouse models to enable translation of these findings into clinical studies. This statement was added at the end of Discussion (lines 476-478):

“To confirm our findings, in vitro and in vivo studies with higher number of patient-derived GSCLCs are planned. These studies are focused on evaluating the antitumor effect of injected super-charged NK cells in GBM-bearing humanized BLT mice.”

Comment: The studies with CSF are not appropriately described in the methods section (ethics, experimental details e.g. vol., normalization, site of tapping etc) and results. It is not mentioned how CSF was tapped, whether it was through a sterile lumbar puncture procedure or whether this fluid was obtained from an intraoperative procedure. The latter can conceivably be associated with blood contamination, a source of error that may affect the high numbers and type of immune cells detected in this CSF. More information is required

Reply: *We apologize for not including the method of CSF collection and immune cell characterization. We added the method in the Supplementary file to the legend to Supplementary Figure 7, pages 8-9:*

“The study as well as the procedures were approved by the UCLA Institutional Review Board (IRB#11-000781), and all participants signed written informed consent in accordance with the Declaration of Helsinki. Patient’s physician performed the procedure of CSF tapping by lumbar puncture according to the clinical protocol in a hospital. A total volume of 10 mL of CSF was collected in a tube without any medium and processed within 1-3 hours from collection to minimize cell loss. There were no signs of peripheral blood contamination in CSF in the test tube, when CSF arrived in the laboratory. CSF was utilized for flow cytometry analysis. The CSF was spun at 1000 rpm for 5 min, the supernatant fluid was discarded and the cell pellet was suspended in 1x PBS and stained with antibodies against CD45, CD3/CD16/CD56, CD3, CD8, CD4, CD14 and CD19, and analyzed using flow cytometry”.

Comment: Introductory text to NK biology and manuscript as a whole can benefit from greater clarity.

Reply: *We added the text in Introduction to introduce NK cells, their biology and clinical implications in more details (please see lines 69-81, 104-123).*

Besides all these amendments, we want to inform you that some additional changes have been made. Anamarija Habič was added as co-author, because she performed additional experiments. Moreover, 2 additional GBM samples for immunofluorescence in GBM tissue sections have been added to Supplementary Figures 2, 3 and 4.

References

- Barry KC, Hsu J, Broz ML, Cueto FJ, Binnewies M, Combes AJ, Nelson AE, Loo K, Kumar R, Rosenblum MD, Alvarado MD, Wolf DM, Bogunovic D, Bhardwaj N, Daud AI, Ha PK, Ryan WR, Pollack JL, Samad B, Asthana S, Chan V, Krummel MF. A natural killer-dendritic cell axis defines checkpoint therapy-responsive tumor microenvironments. *Nat Med.* 2018 Aug;24(8):1178-1191. doi: 10.1038/s41591-018-0085-8. Epub 2018 Jun 25. PMID: 29942093; PMCID: PMC6475503.
- Biederstädt A, Rezvani K. Engineering the next generation of CAR-NK immunotherapies. *Int J Hematol.* 2021 Nov;114(5):554-571. doi: 10.1007/s12185-021-03209-4. Epub 2021 Aug 28. PMID: 34453686; PMCID: PMC8397867.
- Fritzell S, Sandén E, Eberstål S, Visse E, Darabi A, Siesjö P. Intratumoral temozolomide synergizes with immunotherapy in a T cell-dependent fashion. *Cancer Immunol Immunother.* 2013 Sep;62(9):1463-74. doi: 10.1007/s00262-013-1449-z. Epub 2013 Jun 18. PMID: 23775421.
- Kaur K, Cook J, Park SH, Topchyan P, Kozłowska A, Ohanian N, Fang C, Nishimura I, Jewett A. Novel Strategy to Expand Super-Charged NK Cells with Significant Potential to Lyse and Differentiate Cancer Stem Cells: Differences in NK Expansion and Function between Healthy and Cancer Patients. *Front Immunol.* 2017 Apr 5;8:297. doi: 10.3389/fimmu.2017.00297. PMID: 28424683; PMCID: PMC5380683.
- Kaur K, Ko MW, Ohanian N, Cook J, Jewett A. Osteoclast-expanded super-charged NK-cells preferentially select and expand CD8+ T cells. *Sci Rep.* 2020 Nov 23;10(1):20363. doi: 10.1038/s41598-020-76702-1. PMID: 33230147; PMCID: PMC7683603.
- Kaur K, Kozłowska AK, Topchyan P, Ko MW, Ohanian N, Chiang J, Cook J, Maung PO, Park SH, Cacalano N, Fang C, Jewett A. Probiotic-Treated Super-Charged NK Cells Efficiently Clear Poorly Differentiated Pancreatic Tumors in Hu-BLT Mice. *Cancers (Basel).* 2019 Dec 24;12(1):63. doi: 10.3390/cancers12010063. PMID: 31878338; PMCID: PMC7017229.
- Kaur K, Safaie T, Ko MW, Wang Y, Jewett A. ADCC against MICA/B Is Mediated against Differentiated Oral and Pancreatic and Not Stem-Like/Poorly Differentiated Tumors by the NK Cells; Loss in Cancer Patients due to Down-Modulation of CD16 Receptor. *Cancers (Basel).* 2021 Jan 11;13(2):239. doi: 10.3390/cancers13020239. PMID: 33440654; PMCID: PMC7826810.
- Kozłowska AK, Tseng HC, Kaur K, Topchyan P, Inagaki A, Bui VT, Kasahara N, Cacalano N, Jewett A. Resistance to cytotoxicity and sustained release of interleukin-6 and interleukin-8 in the presence of decreased interferon- γ after differentiation of glioblastoma by human natural killer cells. *Cancer Immunol Immunother.* 2016 Sep;65(9):1085-97. doi: 10.1007/s00262-016-1866-x. Epub 2016 Jul 20. PMID: 27439500; PMCID: PMC4996719.

Myers JA, Miller JS. Exploring the NK cell platform for cancer immunotherapy. *Nat Rev Clin Oncol*. 2021 Feb;18(2):85-100. doi: 10.1038/s41571-020-0426-7. Epub 2020 Sep 15. PMID: 32934330; PMCID: PMC8316981.

Podergajs N, Motaln H, Rajčević U, Verbovšek U, Koršič M, Obad N, Espedal H, Vittori M, Herold-Mende C, Miletic H, Bjerkvig R, Turnšek TL. Transmembrane protein CD9 is glioblastoma biomarker, relevant for maintenance of glioblastoma stem cells. *Oncotarget*. 2016 Jan 5;7(1):593-609. doi: 10.18632/oncotarget.5477. PMID: 26573230; PMCID: PMC4808020.

Shaim H, Shanley M, Basar R, Daher M, Gumin J, Zamler DB, Uprety N, Wang F, Huang Y, Gabrusiewicz K, Miao Q, Dou J, Alsuliman A, Kerbauy LN, Acharya S, Mohanty V, Mendt M, Li S, Lu J, Wei J, Fowlkes NW, Gokdemir E, Ensley EL, Kaplan M, Kassab C, Li L, Ozcan G, Banerjee PP, Shen Y, Gilbert AL, Jones CM, Bdiwi M, Nunez-Cortes AK, Liu E, Yu J, Imahashi N, Muniz-Feliciano L, Li Y, Hu J, Draetta G, Marin D, Yu D, Mielke S, Eyrich M, Champlin RE, Chen K, Lang FF, Shpall EJ, Heimberger AB, Rezvani K. Targeting the α v integrin/TGF- β axis improves natural killer cell function against glioblastoma stem cells. *J Clin Invest*. 2021 Jul 15;131(14):e142116. doi: 10.1172/JCI142116. PMID: 34138753; PMCID: PMC8279586.

Tseng HC, Cacalano N, Jewett A. Split anergized Natural Killer cells halt inflammation by inducing stem cell differentiation, resistance to NK cell cytotoxicity and prevention of cytokine and chemokine secretion. *Oncotarget*. 2015 Apr 20;6(11):8947-59. doi: 10.18632/oncotarget.3250. PMID: 25860927; PMCID: PMC4496194.

Vivier E, Tomasello E, Baratin M, Walzer T, Ugolini S. Functions of natural killer cells. *Nat Immunol*. 2008 May;9(5):503-10. doi: 10.1038/ni1582. PMID: 18425107.

Zhang D, Qiu B, Wang Y, Guan Y, Zhang L, Wu A. Temozolomide increases MHC-I expression via NF- κ B signaling in glioma stem cells. *Cell Biol Int*. 2017 Jun;41(6):680-690. doi: 10.1002/cbin.10773. Epub 2017 May 2. PMID: 28403532